# The French version of the Gilles de la Tourette Syndrome Quality of Life Scale for adolescents (GTS-QOL-French-Ado): Adaptation and psychometric evaluation

Isabelle Jalenques[1]*, Candy Guiguet-Auclair[2], Diane Cyrille[3], Clement Debosque[3], Philippe Derost[4], Andreas Hartmann[5], Sophie Lauron[3], Clara Jameux[3], Urbain Tauveron—Jalenques[3], Fabien Rondepierre[1], for The Syndrome de Gilles de La Tourette Study Group[¶]

1 Service de Psychiatrie de l'Adulte A et Psychologie Médicale, Centre de Compétence Gilles de la Tourette, CNRS, Institut Pascal, CHU Clermont-Ferrand, Université Clermont Auvergne, Clermont-Ferrand, France, 2 CNRS, Institut Pascal, CHU Clermont-Ferrand, Université Clermont Auvergne, Clermont-Ferrand, France, 3 Service de Psychiatrie de l'Adulte A et Psychologie Médicale, CNRS, Institut Pascal, CHU Clermont-Ferrand, Université Clermont Auvergne, Clermont-Ferrand, France, 4 Service de Neurologie, CHU Clermont-Ferrand, Université Clermont Auvergne, Clermont-Ferrand, France, 5 Department of Neurology, National Reference Center for Tourette Syndrome, Groupe Hospitalier Pitié-Salpêtrière, Paris, France

¶ Membership of The Syndrome de Gilles de La Tourette Study Group is provided in the Acknowledgments.
* ijalenques@chu-clermontferrand.fr

## Abstract

### Introduction

The aim of this study was to create a new version of the French GTS-QOL adapted to adolescents with GTS aged 12–16 years (GTS-QOL-French-Ado) and to evaluate its psychometric properties.

### Methods

We assessed the psychometric properties of the GTS-QOL-French-Ado in 84 adolescents (mean age 13.6 years, standard deviation 1.2) in terms of factor structure, internal consistency, reliability and convergent validity with the Child Depression Inventory (CDI), the Multidimensional Anxiety Scale for Children (MASC), the Motor tic, Obsessions and compulsions, Vocal tic Evaluation Survey (MOVES) and the French "Vécu et Santé Perçue de l'Adolescent" (VSP-A), a generic self-administered measure of health-related quality of life (HRQoL) in adolescents.

### Results

Exploratory factor analysis of the GTS-QOL-French-Ado resulted in a 5-factor solution. The GTS-QOL-French-Ado demonstrated good acceptability with missing values per subscale ranging from 0% to 1.2%, good internal consistency for four of the five subscales with Cronbach's alpha ranging from 0.56 to 0.87 and good test–retest reliability with intraclass correlation coefficients ranging from 0.74 (95% CI: 0.52–0.86) to 0.82 (95% CI: 0.66–0.91).

**Data Availability Statement:** All data are available at Mendeley (doi: 10.17632/535ctxxsn5.1).

**Funding:** IJ received a grant from the University Hospital of Clermont-Ferrand (AOI 2012) and the Association Française Syndrome Gilles de la Tourette (AO AFSGT 2015). The funders had no role in study design, data collection and analysis, decision to publish, or preparation of the manuscript.

**Competing interests:** The authors have declared that no competing interests exist.

Convergent validity was supported by correlations with CDI, MASC, MOVES, VSP-A and clinical variables.

## Discussion

The GTS-QOL-French-Ado is the first disease-specific HRQoL tool for French-speaking adolescents with GTS aged 12–16 years, and shows good psychometric properties. Further psychometric testing on responsiveness to change would be of great interest.

## Introduction

The Gilles de la Tourette syndrome (GTS) is a chronic neurodevelopmental disorder characterized by multiple motor and one or more vocal tics persisting for more than a year since the first tic onset, which occurs typically in childhood [1]. Two meta-analyses suggested GTS prevalence rates between 0.6 and 0.8% in children and adolescents [2, 3]. Several studies reported that persons with GTS have a poor health-related quality of life (HRQoL) across the lifespan that is, however, perceived differently by children, adolescents and adults [4]. The optimal assessment and treatment strategy for people with GTS must take into account the severity of the tics and comorbid disorders and their effects on daily functioning and HRQoL [5]. The first relevant studies, irrespective of age group, used various generic HRQoL scales [6–10]. However, disease-specific HRQoL instruments are more sensitive to disease experiences [11]. Thus, the Gilles de la Tourette Syndrome–Quality of Life Scale (GTS-QOL), a self-rated disease-specific questionnaire for adults with GTS was developed and validated in English [12]. The use of the GTS-QOL is advocated in the European Clinical Guidelines for Tourette Syndrome and Other Tic Disorders [13]. Our team had validated a French version of the GTS-QOL for adults [14] that we decided to adapt for French speaking adolescents. Indeed, to our knowledge, there has been only one study in France that has assessed the HRQoL of adolescents with GTS, using the generic HRQoL instrument "Vécu et Santé Perçue de l'Adolescent" (VSP-A) [15]. Nevertheless, the early diagnosis of GTS, the stages of life and development specific to adolescence, necessitate a scale suited specifically to adolescents, especially since studies have highlighted differences in HRQoL domains across different age groups of patients with GTS [4, 16].

So, the aim of this study was to adapt, validate the GTS-QOL, and assess its psychometric properties in adolescents with GTS aged between 12 and 16 years and residing in France. The perception and definition of HRQoL may vary from one population to another or by generation, and so there was no guarantee that the French version of the GTS-QOL for adolescents faithfully reflected the subscales of the questionnaire for adults. We therefore decided to perform a full-scale psychometric assessment and to explore the factorial structure of the GTS-QOL for adolescents with an exploratory factor analysis of the items [17]. Associations between sociodemographic and clinical characteristics and HRQoL as evaluated by the French version of the GTS-QOL for adolescents (GTS-QOL-French-Ado) were also investigated.

## Methods

The study was performed in two stages: adaptation of the French version of the GTS-QOL for adults [14] to adolescents (GTS-QOL-French-Ado), and assessment of the psychometric properties of the GTS-QOL-French-Ado.

## Phase 1—adaptation of the French version of the adult GTS-QOL to adolescents

The original English language version of the GTS-QOL for adults [12] comprises 27 items relating to the previous 4 weeks that are rated on a 5-point Likert scale, from 0 (no problem) to 4 (extreme problem), grouped into four subscales: 'Psychological' (11 items), 'Physical and activities of daily living' (7 items), 'Obsessive-compulsive' (5 items), and 'Cognitive' (4 items). To calculate subscale scores the individual scores of the items making up the subscale are added up and then normalized to a 0–100 range, with higher scores corresponding to lower HRQoL. A visual analogue scale (VAS) ranging from 0 (representing extreme dissatisfaction with life) to 100 (extreme satisfaction) is also included in the questionnaire. This original version has good psychometric properties [12].

Our team translated in French and cross-culturally adapted the GTS-QOL in adults (GTS-QOL-French). The French version had satisfactory acceptability, internal consistency, convergent validity and reliability [14]. It was immediately adapted for adolescents aged between 12 and 16 years by three clinicians with experience in the management of adolescents with GTS who suggested how to simplify or rephrase items that they thought could be confusing for subjects in this age group. To check the French-speaking adolescents' understanding and interpretation of the items and instructions, the consensus French version was pre-tested on 11 adolescents with GTS. Their responses and comments were assessed and showed that content validity was good and all items were properly understood. This version was therefore adopted by the expert committee as the pre-final adaptation for adolescents (S1 File). We named this version the GTS-QOL-French-Ado. Finally, the psychometric properties of the GTS-QOL-French-Ado were examined in an independent sample of adolescents with GTS.

## Phase 2—evaluation of the psychometric properties

**Study design.** The study was given approval by the French regional ethics committee "Comité d'Ethique des Centres d'Investigation Clinique de l'inter-région Rhône-Alpes-Auvergne–CE-CIC Grenoble" (n° IRB 0005921, 20 September 2012). All adolescents aged between 12 and 16 years old who accepted to participate and their parents received clear information on the aims and procedures of the study and gave written informed consent.

Participants were given a set of questionnaires during a routine consultation or received them by mail with a return envelope. To assess the test-retest reliability of the GTS-QOL-French-Ado, a subsample of participants (selected by simple random sampling using random number tables) were mailed the questionnaire a second time 15 days after the first assessment. Respondents who reported any degradation in their health status between test and retest were excluded from the reliability analysis [18].

The sample size of the study was determined on the basis of quality criteria established by COSMIN [18, 19] and Terwee et al. (2007) [20]. A sample size of more than 50 subjects is rated good by the COSMIN group for the internal consistency evaluation. A minimum number of 100 subjects or six times the number of items is recommended to ensure satisfactory factor analysis. A sample size of at least 50 subjects is recommended to guarantee acceptable assessment for reliability [20].

**Participants.** Participants were recruited from two French specialist centres [GTS Reference Centre (Paris), GTS Competence Centre (Clermont- Ferrand)] and the "Association Française Syndrome Gilles de la Tourette". They fulfilled DSM-IV-TR criteria for the diagnosis of GTS. Those who consented to participate and who had a postal address then received a set of questionnaires to self-complete.

**Study variables and instruments.** The French versions of the GST-QOL for adolescents, Child Depression Inventory (CDI) [21], Multidimensional Anxiety Scale for Children (MASC) [22], Motor tic, Obsessions and compulsions, Vocal tic Evaluation Survey (MOVES) [23] and "Vécu et Santé Perçue de l'Adolescent" questionnaire (VSP-A) [24] were self-administered.

The CDI is a 27-item self-administered instrument that measures depression symptoms (including cognitive, affective, somatic and behavioural aspects) in children and adolescents aged 7–17 years [25]. Each item is scored 0 (symptom not present), 1 (symptom present and mild) or 2 (symptom present and marked), to obtain a total score ranging from 0 to 54.

The MASC is a self-report scale assessing anxiety symptoms in children and adolescents aged 8–19 years [26]. It comprises 39 items, scored on a 4-point Likert scale from 0 (symptom never applies) to 3 (symptom applies often) and grouped in four subscales: 'Physical Symptoms', 'Social Anxiety', 'Separation Anxiety' and 'Harm Avoidance'. A total score is obtained by adding up the scores for each item.

The MOVES is a self-report scale that measures the severity of tics and other manifestations observed in GTS [27]. It comprises 20 items divided into five subscales, 'Motor tics', 'Vocal tics', 'Obsessions', 'Compulsions' and 'Other associated symptoms' such as copro-, pali- and echophenomena. The score for each subscale is obtained by adding up the scores for the items listed in the subscale. A total score is calculated by adding up the scores of the five subscales, which range from 0 (no symptom) to 60 (the worst condition). For clinical scoring, the 'Motor tics' and 'Vocal tics' scores are combined to obtain a 'Tics' subscale score. The 'Obsessions' and 'Compulsions' scores are added together to form an 'Obsessive-compulsive' subscale score. The MOVES is suggested as a severity scale for tics and related sensory phenomena and recommended as a screening instrument by the Committee on Rating Scale Development of the International Parkinson's Disease and Movement Disorder Society [28].

The VSP-A is a generic self-report instrument assessing adolescents' HRQoL that comprises 39 items of which 34 are divided into 9 subscales: 'Vitality', 'Psychological well-being', 'Relationship with friends', 'Leisure activities', 'Relationship with parents', 'Physical well-being', 'Relationship with teachers', 'School performance' and 'Body image' [24]. Total subscale scores range from 0 to100, with higher scores indicating better HRQoL.

Sociodemographic data (age, gender, way of life, level of education, educational protective measures) and medical data (age at diagnosis and at onset of symptoms, location of tics, medical monitoring, treatment, comorbidities) were also collected. All data are available at Mendeley [29].

**Statistical analysis.** Statistical analyses were performed with SAS v9.4 software (SAS Institute) and conducted at a two-sided alpha = 0.05 significance levelto test the null hypothesis of zero correlation between two variables.

*Data completeness.* Respondent acceptability was determined by the frequency of missing values in the questionnaire.

*Factor analysis.* Factor analysis using the principal axis extraction method and oblique oblimin rotation, which allows correlation of the factors, were performed to study the multidimensionality and distribution of the subscale items [30]. The Kaiser-Meyer-Olkin (KMO) statistic and Bartlett's test of sphericity were used to check the appropriateness of running the factor analysis. KMO values higher than 0.5 are acceptable [31]. Bartlett's test requires to yield significant results (p<0.05). Eigenvalues higher than 1 (Kaiser criterion), Cattell's scree plot [32] and interpretability of factors were used for factor retention. The solution that gave the most appropriate factor structure (item loadings greater than 0.32, no or few item cross loadings, i.e. no or few items with loadings at 0.32 or higher on two or more factors) was adopted [30].

*Descriptive statistics and score distributions.* Mean, standard deviation, median, range and coefficient of skewness were used. The floor and ceiling effects were used to study variability in the GTS-QOL-French-Ado scores for each subscale. The effects were considered to occur if more than 15% of the participants obtained the lowest or highest possible score [33].

*Internal consistency.* The internal consistency of each subscale was assessed by Cronbach's α [34] and McDonalds' omega coefficients [35]. The minimum required for these coefficients was 0.70, according to the standard used for group comparisons [36, 37].

*Item-total correlations.* Item-total consistency was used to assess the extent of the linear relationship between an item and its subscale, corrected for overlap (the item to be correlated with the scale was omitted from the scale total) [38]. A minimum correlation coefficient of 0.40 was considered to be indicative of good item-total consistency [39].

*Inter-subscale correlations.* Spearman's coefficients were used to evaluate inter-subscale correlations. Correlations were considered to be very small for coefficients lower than 0.30, small for coefficients between 0.30 and 0.50, moderate from 0.50 to 0.70 and strong if higher than 0.70 [40].

*Reliability.* The test-retest method was used to assess stability over time. Reliability of the subscales was estimated by intraclass correlation coefficient (ICC, absolute agreement, two-way mixed effect model with single measurement) [41]. Coefficients higher than 0.70 were considered to be satisfactory [20].

*Convergent validity.* The relationships between GTS-QOL-French-Ado subscale scores and those of (1) CDI, (2) MASC, (3) MOVES and (4) VSP-A were studied by calculating Spearman ρ correlation coefficients. Positive correlations were expected between CDI, MASC, MOVES subscale scores and those of the GTS-QOL-French-Ado because low scores indicated good conditions in all four questionnaires. Negative correlations were expected between GTQ-QOL-French-Ado and VSP-A subscale scores since low scores indicated good conditions for the GST-QOL-French-Ado but bad conditions for the VSP-A. Correlations were considered very small for coefficients lower than 0.30, small for coefficients between 0.30 and 0.50, moderate from 0.50 to 0.70 and strong if higher than 0.70 [40].

*Associations between sociodemographic and clinical characteristics and HRQoL.* The GTS-QOL-French-Ado subscale scores were compared according to age, gender, disease duration, vocal tics, number of treatments for GTS, medical monitoring and comorbidities. Effect sizes (ES) were calculated by Hedges' g (a variation of Cohen's d ES that corrects for bias due to small sample sizes) and its 95% confidence interval [42]. Absolute values of 0.20 or more commonly indicated a small ES, 0.50 or more, moderate ES, and 0.80 or more, large ES [43]. For all subscales of GTS-QOL-French-Ado, a high positive ES indicated a significant negative impact on HRQoL.

## Results

### Participants

Of the 123 questionnaires sent to those who had agreed to participate in the study, 84 (68.3%) were returned. The sociodemographic and clinical characteristics of the participants are given in Table 1. They were predominantly male (80.5%) with a mean age of 13.6 years (SD 1.2) and a median age of 13.5 years (interquartile range 12.4–14.5). In 80.7% of cases (n = 67) they lived with both parents, in 15.7% (n = 13) with one of the parents, in 2.4% (n = 2) with another family member, and in one case (1.2%) in an institution. First symptoms began 7.5 years (SD 2.7) before the study and diagnosis of GTS had been made on average 3.8 years before (SD 2.5). Most declared they had had tics in the last month (95.2%), current medical monitoring

**Table 1. Sociodemographic and clinical characteristics of the participants.**

|  | n = 84 |
|---|---|
| Age (years), *mean (SD)* | 13.6 (1.2) |
| Male, *n (%)* | 66 (80.5) |
| Level of education, *n (%)* |  |
| Primary school | 1 (1.2) |
| Secondary school | 71 (85.5) |
| High school | 11 (13.3) |
| Educational protective measures, *n (%)* | 11 (13.3) |
| Age at diagnosis of GTS (years), *mean (SD)* | 9.8 (2.5) |
| Age at the first symptoms (years), *mean (SD)* | 6.1 (2.4) |
| Tics as first symptoms, *n (%)* | 68 (82.9) |
| Location of first tics [a], *n (%)* |  |
| Face | 54 (65.1) |
| Neck | 19 (22.9) |
| Trunk | 4 (4.8) |
| Shoulder | 16 (19.3) |
| Upper limbs | 18 (21.7) |
| Lower limbs | 7 (8.4) |
| Vocal tics | 51 (61.4) |
| Others | 7 (8.4) |
| Tics in the last month, *n (%)* | 79 (95.2) |
| Current motor tics, *n (%)* | 75 (90.4) |
| Current vocal tics, *n (%)* | 66 (79.5) |
| Frequency of current tics, *n (%)* |  |
| Several per minute | 22 (27.8) |
| Several per hour | 14 (17.7) |
| Several per day | 11 (13.9) |
| Very variable | 32 (40.5) |
| Medical monitoring for GTS [a], *n (%)* | 79 (94.0) |
| General practitioner | 21 (25.3) |
| Paediatrician | 2 (2.4) |
| Neurologist | 62 (74.7) |
| Psychiatrist or child psychiatrist | 33 (39.8) |
| Psychologist | 30 (36.1) |
| Multidisciplinary consultation [b] | 34 (41.0) |
| Other monitoring [a], *n (%)* | 14 (16.9) |
| Psychomotor therapist | 6 (7.1) |
| Physiotherapist | 3 (3.6) |
| Speech and language therapist | 3 (3.6) |
| Occupational therapist | 1 (1.2) |
| Psychotherapy for psychological suffering | 27 (34.6) |
| Treatment reported for tics [a], *n (%)* | 63 (75.0) |
| Neuroleptics/Antipsychotics [c] | 61 (72.6) |
| First generation | 4 (4.8) |
| Second generation | 50 (59.5) |
| First and second generation | 7 (8.3) |
| Others | 10 (11.9) |
| Clonidine | 3 (3.6) |

(*Continued*)

**Table 1.** (Continued)

|  | **n = 84** |
|---|---|
| Antiepileptics | 6 (7.1) |
| Baclofen | 1 (1.2) |
| Naltrexone | 1 (1.2) |
| Other treatment reported [a], *n (%)* | 28 (33.3) |
| Attention-deficit/hyperactivity disorders | 13 (15.5) |
| Antidepressants | 12 (14.3) |
| Anxiolytics | 5 (6.0) |
| Sleep disorders | 1 (1.2) |
| Morphine sulfate | 1 (1.2) |
| Comorbidities [c], *n (%)* | 35 (42.2) |
| Mental and behavioural disorders [a], *n (%)* | 32 (38.6) |

[a] An adolescent could have a different location for first tics, different medical monitoring, different treatment and different mental or behavioural disorders.

[b] Neurologist and psychiatrist (*n = 11*), neurologist and psychologist (*n = 13*), neurologist and psychiatrist and psychologist (*n = 10*).

(94.0%) and current treatment for GTS (82.1%). Mental and behavioural disorders were the most common comorbidities, in particular attention-deficit/hyperactivity disorders (n = 19, 22.6%), anxiety/depressive disorders (n = 12, 14.3%) and specific developmental disorders of scholastic skills (n = 5, 6.0%).

Depression, anxiety, severity of tics and other symptoms, and HRQoL based on responses to CDI, MASC, MOVES and VSP-A questionnaires, respectively, are given in Table 2.

## Data completeness

The proportion of missing values per item was low and ranged from 0% to 2.4%. However, 41 participants (49.4%) declared they received help (reading or writing) to fill in the questionnaire from their mother (92.7%), father (4.9%) or both (2.4%).

## Factor analysis

KMO measure of sampling adequacy was 0.795 and the significance value of Bartlett's test of sphericity was <0.0001 ($\chi^2$ = 1038.4), indicating that the data were suitable for factor analysis. Factor analysis (oblimin rotation) of the 27 items of the GTS-QOL-French-Ado identified five interpretable factors with eigenvalues higher than one (S1 Fig), which accounted for 59.0% of the total variance (Fig 1). The five subscales identified were designated: 'Psychological' (9 items), 'Social' (6 items), 'Echo-coprophenomena/Obsessive-compulsive' (5 items), 'Cognitive' (4 items) and 'Physical' (3 items). All items loaded higher than 0.32 on their subscale. Five items loaded at 0.32 or higher on two factors: item 9 on factors 1 and 5 (0.40 and 0.39, respectively); items 15 and 16 loaded on factors 1 and 2 (0.41 and 0.41, respectively, for item 15, 0.36 and 0.57 for item 16); item 8 on factors 3 and 4 (0.66 and 0.34, respectively); and item 11 on factors 4 and 5 (0.32 and 0.39, respectively) (S1 Table). After assessment of internal consistency and item-total correlations and taking into account the clinical relevance of the factor structure, all but item 11 were conserved in the subscale where they loaded highest. Item 11 was kept in factor 4 to increase the clinical relevance and internal consistency of the subscale.

**Table 2. Depression (CDI instrument), anxiety (MASC instrument), severity of tics (MOVES instrument) and health-related quality of life assessed by a generic instrument (VSP-A).**

|  | Mean (SD) | Range possible |
|---|---|---|
| CDI total score [a] | 13.0 (7.6) | 0–54 |
| MASC scores [a] |  |  |
| Physical symptoms | 10.7 (5.5) | 0–27 |
| Social anxiety | 15.0 (7.2) | 0–30 |
| Separation anxiety | 9.3 (6.8) | 0–33 |
| Harm avoidance | 15.5 (4.7) | 0–27 |
| Total | 50.7 (20.5) | 0–117 |
| MOVES scores [a] |  |  |
| Motor tics | 6.2 (3.2) | 0–12 |
| Vocal tics | 3.8 (2.7) | 0–12 |
| Tics | 10.1 (4.8) | 0–24 |
| Obsessions | 3.1 (3.1) | 0–12 |
| Compulsions | 4.4 (2.8) | 0–12 |
| Obsessive-compulsive | 7.5 (5.3) | 0–24 |
| Other associated symptoms | 1.9 (2.2) | 0–12 |
| Total | 19.6 (10.8) | 0–60 |
| VSP-A scores [b] |  |  |
| Vitality | 59.9 (23.0) | 0–100 |
| Psychological well-being | 56.4 (25.4) | 0–100 |
| Relationship with friends | 55.7 (25.5) | 0–100 |
| Leisure activities | 45.0 (25.3) | 0–100 |
| Relationship with parents | 67.4 (21.6) | 0–100 |
| Physical well-being | 60.6 (22.6) | 0–100 |
| Relationship with teachers | 65.7 (26.0) | 0–100 |
| School performance | 58.6 (28.2) | 0–100 |
| Body image | 66.1 (35.4) | 0–100 |

[a] Worse conditions indicated by higher scores.

[b] Worse conditions indicated by lower scores.

## Descriptive statistics, score distribution, floor and ceiling effects

Table 3 shows the descriptive statistics and score distributions for the GTS-QOL-French-Ado subscales. No floor or ceiling effects were found. Lower scores, indicative of better HRQoL, were observed for the 'Social' and 'Echo-coprophenomena/Obsessive-compulsive' subscales. Coefficients of skewness were the highest for these subscales, with a positive distribution towards good health. Higher scores, corresponding to lower HRQoL, were found for the 'Psychological' and 'Cognitive' subscales.

## Internal consistency

The GTS-QOL-French-Ado subscales showed good internal consistency, except for the 'Physical' subscale (Table 4). Cronbach's α and McDonalds' omega coefficients were similar for the 'Psychological' and 'Social' subscales with values of 0.87 and 0.86 respectively for these subscales. Cronbach's α and McDonalds' omega coefficients were equal to 0.81 and 0.80 respectively for the 'Echo-coprophenomena/Obsessive-compulsive' subscale. The 'Cognitive' and 'Physical' subscales did not obtain the minimum required coefficients of 0.70. Nevertheless,

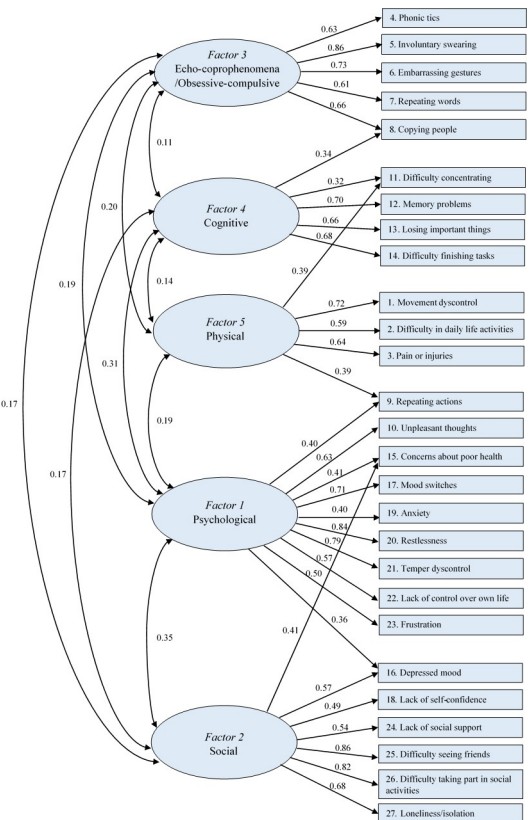

**Fig 1. Path diagram of the factorial structure of the GTS-QOL-French-Ado from the exploratory factor analysis.**
Factor latent variables are represented by ellipses. Measured variables (items of the GTS-QOL-French-Ado) are
represented by rectangles. The curved double-headed links represent the correlations among factors. The directed
links from factors to items represent the factor loadings estimates. The path diagram displays only the links that have
loadings equal or greater than 0.32.

the 'Cognitive' subscale had very close coefficients, with a Cronbach's α of 0.68 and a McDo-
nalds' omega of 0.69. The 'Physical' subscale obtained a Cronbach's α of 0.56 and a McDo-
nalds' omega of 0.57.

## Item-total correlations

Almost all corrected item-total correlations were higher than the required value of 0.40, and
ranged from 0.31 to 0.74, which is indicative of good item-total consistency (S2 Table). They
were lower for item 9 of the 'Psychological' subscale (value of 0.35), for item 13 of the

**Table 3. Descriptive statistics and score distributions of the GTS-QOL-French-Ado subscales.**

| GTS-QOL-French-Ado subscales | Missing values (%) | Mean (SD) | Range | Median | Coefficient of skewness | Floor effect (%) | Ceiling effect (%) |
|---|---|---|---|---|---|---|---|
| Psychological | 0 | 40.9 (25.9) | 0–100 | 38.9 | 0.4 | 3.6 | 0 |
| Social | 1.2 | 31.5 (28.2) | 0–100 | 25.0 | 1.1 | 7.2 | 3.6 |
| Echo-coprophenomena/Obsessive-compulsive | 1.2 | 25.5 (25.5) | 0–100 | 15.0 | 1.4 | 10.8 | 2.4 |
| Cognitive | 0 | 40.5 (23.7) | 0–100 | 37.5 | 0.4 | 2.4 | 1.2 |
| Physical | 0 | 34.8 (23.9) | 0–100 | 33.3 | 0.4 | 10.7 | 1.2 |

Higher scores indicate worse health-related quality of life.

**Table 4. Internal consistency, inter-subscale correlations and test-retest reliability for the GTS-QOL-French-Ado subscales.**

| GTS-QOL-French-Ado subscales | Psychological | Social | Echo-coprophenomena / Obsessive-compulsive | Cognitive | Physical | ICC (95% CI) |
|---|---|---|---|---|---|---|
| Psychological | **0.87** | | | | | 0.78 (0.58–0.89) |
| Social | 0.66 ** | **0.86** | | | | 0.74 (0.52–0.86) |
| Echo-coprophenomena / Obsessive-compulsive | 0.43 ** | 0.38 ** | **0.81** | | | 0.80 (0.63–0.90) |
| Cognitive | 0.56 ** | 0.58 ** | 0.31 * | **0.68** | | 0.82 (0.66–0.91) |
| Physical | 0.34 * | 0.43 ** | 0.47 ** | 0.34 * | **0.56** | 0.79 (0.61–0.89 |

Cronbach's α are reported on the diagonal and in bold text.

Inter-scale correlations are Spearman's coefficients. Correlations significantly different from zero:

* p<0.01 and

** p<0.001.

ICC (95% CI): intraclass correlation coefficient (95% confidence interval).

'Cognitive' subscale (value of 0.39), and items 2 and 3 of the 'Physical' subscale (values of 0.31 and 0.39 respectively) with values however close to that of 0.40.

In addition, all but 4 items correlated higher with their parent subscale (corrected for overlap) than with the other subscales (S2 Table). For item 19 (anxiety), the item-total correlation with its subscale was 0.54 and the correlation with the 'Social' subscale was 0.58. For item 11 (difficulty concentrating), the item-total correlation with its subscale was 0.42 and correlations with 'Psychological' and 'Social' subscales were 0.49 and 0.46, respectively. For item 14 (difficulty finishing tasks), the item-total correlation with its subscale was 0.48 and the correlation with the 'Social' subscale was 0.50. For item 2 (difficulty in daily life activities), the item-total correlation with its subscale was 0.31 and correlations with 'Social', 'Echo-coprophenomena/ Obsessive-compulsive' and 'Cognitive' subscales were 0.39, 0.41 and 0.32, respectively.

## Inter-subscale correlations

Correlations between all GTS-QOL-French-Ado subscales were positive and all significant, ranging from 0.31 to 0.66 (Table 4). The 'Psychological' subscale had moderate correlations with 'Social' and 'Cognitive' subscales (r = 0.66 and 0.56, respectively) and small correlations with the other subscales (r = 0.43 and 0.34 with 'Echo-coprophenomena/Obsessive-compulsive' and 'Physical' subscales, respectively). The 'Social' subscale had small correlations with 'Echo-coprophenomena/Obsessive-compulsive' and 'Physical' subscales (r = 0.38 and 0.43, respectively) and a moderate correlation with the 'Cognitive' subscale (r = 0.58). The 'Echo-coprophenomena/Obsessive-compulsive' subscale had small correlations with 'Cognitive' and 'Physical' subscales (r = 0.31 and 0.47, respectively). The correlation between 'Cognitive' and 'Physical' subscales was small (r = 0.34).

## Reliability

For the reliability assessment, a subsample of respondents was first invited to complete a second time the GTS-QOL-French-Ado questionnaire. The response rate was very low and, in consequence, all adolescents subsequently included were invited to complete a second time the questionnaire. Thus, the retest was sent to 46 participants. Of the 36 (78.3%) who returned completed questionnaires within 15 days, four reported that their health status had degraded between test and retest. Were retained for test-retest reliability analysis 32 respondents, male

in majority (74.3%) with a mean age of 13.4 years (SD 1.3) and a median age of 13 years (inter-quartile range 12.2–14.2).

Test-retest reliability was good for all GTS-QOL-French-Ado subscales, with ICCs values higher than 0.70, ranging from 0.74 to 0.82 (Table 4).

## Convergent validity

As expected, correlations between GTS-QOL-French-Ado subscales and those of CDI, MASC and MOVES were positive (100% of hypotheses confirmed) (Fig 2).

All correlations between GTS-QOL-French-Ado subscales and CDI total score were significant. Moderate and strong correlations were found between CDI and the 'Psychological' and 'Social' subscales, respectively (r = 0.61 and 0.73). Very small correlations were found between CDI and 'Echo-coprophenomena/Obsessive-compulsive' and 'Physical' subscales (r = 0.28 and 0.26, respectively). A small correlation was found between CDI and the 'Cognitive' subscale (r = 0.46).

Correlations between GTS-QOL-French-Ado subscales and the MASC total score were all significant. They were moderate with GTS-QOL-French-Ado 'Psychological' and 'Social' subscales (r = 0.54 and 0.58, respectively), small with 'Echo-coprophenomena/Obsessive-compulsive' and 'Physical' subscales (r = 0.35 and 0.39, respectively) and very small with the 'Cognitive' subscale (r = 0.23). Moderate correlations were found between GTS-QOL-French-Ado 'Psychological' subscale and MASC 'Physical symptoms', and 'Separation Anxiety' subscales (r = 0.51and 0.53, respectively). Correlations between the GTS-QOL-French-Ado 'Social' subscale and MASC 'Physical symptoms' (r = 0.50), 'Social Anxiety' (r = 0.61) and 'Separation Anxiety' (r = 0.52) subscales were moderate.

Results showed a small correlation between the total MOVES score and the 'Cognitive' subscale (r = 0.47), and moderate correlations with the other GTS-QOL-French-Ado subscales, but with all being significant (ranging from 0.52 to 0.63). Significant moderate correlations were found between the GTS-QOL-French-Ado 'Psychological' subscale and MOVES 'Vocal

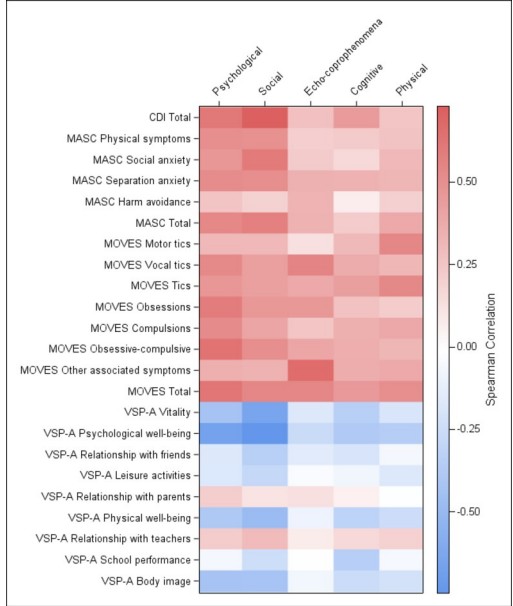

**Fig 2. Convergent validity for GTS-QOL-French-Ado.** Correlations represented on a HEATMAP: in red, positive correlations and in blue, negative correlations. Graduation in the colour indicates the strength of the correlation.

tics' (r = 0.54), 'Obsessions' (r = 0.60), 'Compulsion' (r = 0.53) and 'Obsessive-compulsive' (r = 0.64) subscales; between the GTS-QOL-French-Ado 'Social' and MOVES 'Obsessive-compulsive' subscales (r = 0.51); between the GTS-QOL-French-Ado 'Echo-coprophenomena/Obsessive-compulsive' subscale and MOVES 'Vocal tics' (r = 0.56) and 'Other associated symptoms' (r = 0.67) subscales; and between the GTS-QOL-French-Ado 'Physical' subscale and MOVES 'Motor tics' (r = 0.55) and 'Tics' (r = 0.55) subscales. Small correlations were found between the GTS-QOL-French-Ado 'Cognitive' subscale and MOVES subscales, ranging from 0.28 to 0.44.

The correlations between GTS-QOL-French-Ado and VSP-A subscales were negative, as expected, except for VSP-A 'Relationship with parents' and 'Relationship with teachers' subscales, for which correlations were positive (80% of hypotheses confirmed). The positive significant correlations were very small between these VSP-A subscales and the GST-QOL-French-Ado 'Psychological' subscale (r = 0.23 for the two subscales) and small between the VSP-A 'Relationship with teachers' and the GST-QOL-French-Ado 'Social' subscales (r = 0.31). The VSP-A 'Psychological well-being' subscale correlated moderately with the GTS-QOL-French-Ado 'Psychological' subscale (r = -0.67) and strongly with the 'Social' subscale (r = -0.75). The correlation between the GTS-QOL-French-Ado 'Social' and the VSP-A 'Vitality' subscales was moderate (r = -0.65). Other negative significant correlation coefficients ranged from -0.22 to -0.48, with the lowest ones being observed for the GTS-QOL-French-Ado 'Echo-coprophenomena/Obsessive-compulsive' subscale.

## Association between sociodemographic and clinical characteristics and GTS-QOL-French-Ado subscale scores

No significant correlation was observed between GTS-QOL-French-Ado subscale scores and disease duration. A significant but very small correlation was found between the 'Social' subscale and age (r = 0.25). GTS-QOL-French-Ado subscale scores were not significantly different according to gender and comorbidities.

The number of current treatments for GTS was significantly correlated with all subscales, the higher the number of treatments the poorer the HRQoL: r = 0.29 with 'Psychological' subscale, r = 0.44 with the 'Social' subscale, r = 0.23 with the 'Echo-coprophenomena/Obsessive-compulsive' subscale, r = 0.35 with the 'Cognitive' subscale and r = 0.36 with the 'Physical' subscale.

The effect size of current vocal tics and medical monitoring on GTS-QOL-French-Ado subscales are showed in Fig 3.

Current vocal tics were significantly associated with worse HRQoL for the 'Echo-coprophenomena/Obsessive-compulsive' subscale, with a moderate ES.

Participants with multidisciplinary medical monitoring had a significantly worse HRQoL, except for the 'Echo-coprophenomena/Obsessive-compulsive' subscale, with moderate ES, the highest being for the 'Psychological' and 'Social' subscales.

Participants with medical monitoring provided by a psychiatrist or child psychiatrist had a significantly poorer HRQoL for the 'Psychological' subscale with a moderate ES. Medical monitoring by a psychologist was significantly associated with worse HRQoL for the 'Psychological' and 'Cognitive' subscales, with moderate ES.

Psychotherapy for treatment of tics was significantly associated with poorer HRQoL for the 'Psychological', 'Social', and 'Echo-coprophenomena/Obsessive-compulsive' subscales, with moderate ES. Psychotherapy for psychological suffering was significantly associated with worse HRQoL for the 'Psychological', 'Social' and 'Cognitive' subscales, with large ES.

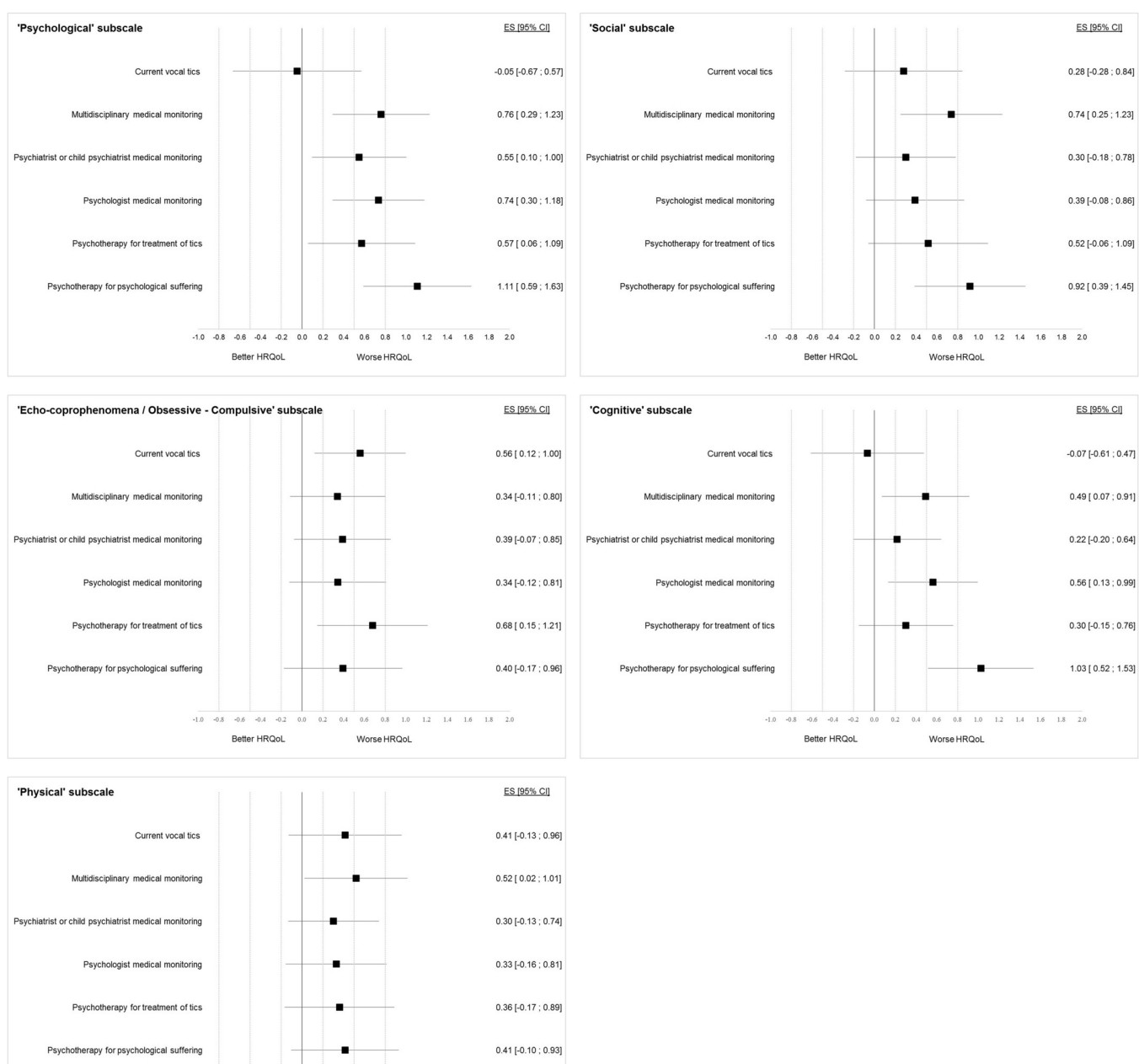

**Fig 3. Effect size of sociodemographic and clinical factors on GTS-QOL-French-Ado subscales.** Effect sizes (ES) are presented as forest plot with 95% confidence interval [95% CI] for each GTS-QOL-French-Ado subscale. The ES are given for 'Yes' versus 'No' values of the sociodemographic and clinical factors. Positive ES indicates worse HRQoL and negative ES better HRQoL. Dotted lines represent the threshold for small (0.2), moderate (0.5) and large (0.8) ES.

## Discussion

Our study described the evaluation of the psychometric properties of the GTS-QOL-French-Ado. It is also the first study to use a disease-specific questionnaire to assess the HRQoL of a sample of adolescents with GTS residing in France.

The adolescents with GTS who took part in this study were similar to those described elsewhere in terms of male-female ratio and age at first symptoms [1, 44]. The study population was representative owing to diversified recruitment, not only from a tertiary but also from a secondary referral centre and from a patient association, which is a strength of our study. The comprehensibility of the scale items was judged on the basis of a clinical sample, which however was small.

The GTS-QOL-French-Ado had good acceptability with very low percentages of missing values and good response distribution, indicating that the questionnaire is adapted to the target population. However, 41 of the participants in the study declared having received help (reading or writing) to fill in the questionnaire, mainly from their mothers. Nevertheless, they were not different from the others in terms of age, frequency and severity of current tics, and mental and behavioural disorders.

In parallel of our work (that began in 2012), two Italian and English versions of the GTS-QOL for children and adolescents, named the C&A-GTS-QOL, were developed and validated in 2013 and 2016 [45, 46]. The five-factor structure of the GTS-QOL-French-Ado differs from the four-factor structure of the Italian version and the six-factor structure of the English-language version of the C&A-GTS-QOL [45, 46] and from the six-factor structure of the GTS-QOL-French for adults [14]. Compared to the Italian version [45], the GTS-QOL-French-Ado has one additional subscale, the 'Social' subscale, and the other four subscales are not composed of exactly the same items. Compared to the English-language version of the C&A-GTS-QOL [46], the GTS-QOL-French-Ado does not have an 'ADL' subscale, and obsessive-compulsive symptoms and echo-coprophenomena are grouped together in one subscale. Compared to the GTS-QOL-French for adults, the 'Psychological' subscale comprises the three items of the 'Obsessive-compulsive' subscale. A 'Psychological' subscale exists in all versions. Apart from the English-language version, which is a little different, this factor contains the largest number of items and explains a large part of the variance (31.4%) in the GTS-QOL-French-Ado, as in the GTS-QOL-French for adults (41.2%) and the Italian version of the C&A-GTS-QOL (34.0%). This is perfectly consistent with the importance of this domain of the self-reported disease-specific quality of life in GTS [4]. Item 16 (Depressed mood) overlaps between the 'Psychological' and 'Social' subscales in the GTS-QOL-French-Ado, we chose to place it in the 'Social' subscale, enhancing the psychometric properties of the subscale. Our choice is consistent with recent results concerning associations of loneliness and social isolation with depressive symptoms among adolescents [47, 48]. Moreover, in the Italian and English versions of the C&A-GTS-QOL, item 16 loaded on a psychological factor that comprised items of the GTS-QOL-French-Ado 'Social' subscale.

Unlike in the Italian and English-language versions of the C&A-GTS-QOL [45, 46], the 'Psychological' subscale of the GTS-QOL-French-Ado does not include items related to social domain grouped together in the 'Social' subscale. A 'Social' subscale also exists in the GTS-QOL-French for adults [14]. Several studies found impaired social functioning and peer relationship problems in youths with GTS [4, 49, 50]. The emergence of this 'Social' subscale could be of great use to specifically assess the social dimension of HRQoL.

The 'Echo-coprophenomena/Obsessive-compulsive' subscale is close to that of the GTS-QOL-French for adults. Item 11 (Difficulty concentrating) overlaps between the 'Cognitive' and 'Physical' subscales in the GTS-QOL-French-Ado. We opted to place this item in the 'Cognitive' subscale, because it is relevant from a clinical and psychometrical point of view. The 'Cognitive' subscale is identical in the different versions for adults and children/adolescents of the GTS-QOL (except for the English-language C&A-GTS-QOL, in which the 'Cognitive' subscale also includes items generally classified in the 'Psychological' dimension). This subscale explains the fairly similar proportion of the variance (about 6%) in all these studies. It

also suggests that the HRQoL of adolescents with GTS could be markedly affected by cognitive factors, which had not been evidenced so far in adolescents.

We performed an exploratory factor analysis (EFA) instead of a confirmatory factor analysis (CFA), to explore the possible underlying factor structure of the GTS-QOL-French-Ado without imposing a preconceived structure on the outcome. In the framework of a CFA, a hypothesized model is established and the CFA confirm that it provides a good fit to the observed data. This makes CFA a method suited for validation of instruments with a predefined measurement model. However, given previous studies on the GTS-QOL for adults [12, 14] or adapted for adolescents [45, 46] that found different factor structures, such theoretical model seems lacking. Moreover, the perception and definition of HRQoL may vary from adults to adolescents and could impact the factorial structure of the scale. Another reason to not have performed a CFA is the sample size of our study, which is a major inconvenience to conduct a CFA. Recommendations for the sample size when conducting a CFA are disparate, ranging from 150 to 1000 subjects [51], and the large sample size is a more crucial requirement in CFA compared to EFA [52].

The GTS-QOL-French-Ado subscales showed good internal consistency, except for the 'Physical' subscale. In the English version of the C&A-GTS-QOL, this dimension showed also moderate internal consistency with a Cronbach' alpha coefficient value of 0.50 [46]. The item-total consistency was good. The inter-subscale correlations were satisfactory. The test-retest reliability, which is an essential property [53], showed good reliability for all subscales. This property was not explored in the Italian and English versions of the C&A-GTS-QOL [45, 46].

Lastly, the GTS-QOL-French-Ado had good convergent validity when compared to the CDI, MASC, MOVES, and VSP-A questionnaires (92.2% of hypotheses were met). We found positive correlations between the 'Psychological' subscale and the CDI and MASC total scores evaluating, respectively, depression and anxiety, which have been shown to affect the HRQoL of persons with GTS [4, 10]. Also important are the correlations between the CDI and MASC total scores and the 'Social' subscale of the GTS-QOL-French-Ado, which are consistent with results of previous studies [54, 55]. Su et al. (2017) compared the C&A-GTS-QOL with the Paediatric Quality of Life Inventory 4.0 (PedsQL) [46]. We propose a different approach that shows a good convergent validity of the GTS-QOL-French-Ado with the VSP-A and was used in the first study of the HRQoL of French-speaking adolescents with GTS [15]. We found logical negative strong correlations between the 'Psychological' and 'Social' subscales of the GTS-QOL-French-Ado and the VSP-A 'Psychological well-being' subscale.

The assessment made with the GTS-QOL-French-Ado and the VSP-A showed that the HRQoL of the adolescents with GTS was overall impaired and confirmed the results of the first study made in France [15]. Our findings are also consistent with those of previous studies in other countries [4].

The major limitation of our study concerns the sample size. The recommended sample size for the factor analysis and the reliability test could not be included [18, 19]. We obtained a ratio of 3.1 respondents for each question, which might be a limitation for the factor analysis. The factorial structure of the GTS-QOL-French-Ado showed cross-loadings for 5 items. Neither the English-language nor the Italian C&A-GTS-QOL validations indicated whether items loaded on several subscales [45, 46]. Despite the fact that the retest was proposed to all adolescents included in a second time, the resulted number of participants in the test-retest reliability was lower than the one attempted. Nevertheless, in the adapted COSMIN Risk of Bias checklist [19], no recommendation was made on the sample size required for reliability analysis and good agreement was obtained in our study for all subscales despite the limited number of participants. Further evaluation of the psychometric properties of the GTS-QOL-French-Ado are needed on larger samples to confirm our first results, and notably its five-factor structure.

However, to our knowledge, our study, using a specific HRQoL instrument, is one of those including the higher number of adolescents with GTS [4, 49, 56–60].

The self-completion of questionnaires could be considered as a limitation of our work. We cannot exclude that the rate of some comorbidities was slightly underestimated, nor that adolescents who had high rates of tics or certain comorbidities, especially mental and behavioural disorders, were less actively involved in the study. However, we took care to assess all comorbidities—those declared by the adolescents and those indicated by the treatments they were receiving. Additionally, our wider recruitment is reflected in both the average severity of SGT and the average rate of comorbid mental and behavioural disorders. Finally, we did not study sensitivity to change after treatment.

## Conclusion

The GTS-QOL-French-Ado questionnaire, which has shown satisfactory internal consistency, convergent validity and reliability, is well suited to assess the HRQoL of French-speaking adolescents with GTS. This disease-specific instrument can therefore be preferred to other generic instruments to assess HRQoL in French-speaking adolescents with GTS. Further studies on larger sample to confirm these first results and longitudinal studies to assess responsiveness to change with therapeutic interventions would be of great interest.

## Supporting information

**S1 File. The GTS-QOL-French-Ado.**
(PDF)

**S1 Fig. Scree plot of the eigenvalues for the factor analysis of the GTS-QOL-French-Ado.**
(TIF)

**S1 Table. Factor loadings from the factor analysis of the GTS-QOL-French-Ado items.**
(DOCX)

**S2 Table. Corrected item-total correlations for the GTS-QOL-French-Ado subscales.**
(DOCX)

## Acknowledgments

Members of the The Syndrome de Gilles de La Tourette Study Group are Diane Cyrille, Clement Debosque, Philippe Derost, Loïc Duron, Candy Guiguet-Auclair, Isabelle Jalenques (leader of the group, ijalenques@chu-clermontferrand.fr), Sophie Lauron, Guillaume Legrand and Fabien Rondepierre (CHU Clermont-Ferrand, Clermont-Ferrand, France); Clara Jameux, Urbain Tauveron-Jalenques and Jeffrey Watts (Université Clermont Auvergne, Clermont-Ferrand, France); Andreas Hartmann (Assistance Publique-Hôpitaux de Paris AP-HP, Paris, France),

The authors thank the participants, and J Watts for advice on the English version of the manuscript, and the "Association Française Syndrome Gilles de la Tourette" for its help in the realization of the study.

## Author Contributions

**Conceptualization:** Isabelle Jalenques, Candy Guiguet-Auclair, Philippe Derost, Urbain Tauveron—Jalenques, Fabien Rondepierre.

**Data curation:** Diane Cyrille, Clement Debosque, Clara Jameux.

**Formal analysis:** Candy Guiguet-Auclair, Fabien Rondepierre.

**Funding acquisition:** Isabelle Jalenques.

**Investigation:** Diane Cyrille, Clement Debosque, Clara Jameux.

**Methodology:** Isabelle Jalenques, Candy Guiguet-Auclair, Urbain Tauveron—Jalenques, Fabien Rondepierre.

**Project administration:** Fabien Rondepierre.

**Resources:** Isabelle Jalenques, Philippe Derost, Sophie Lauron, Fabien Rondepierre.

**Software:** Candy Guiguet-Auclair.

**Supervision:** Isabelle Jalenques, Andreas Hartmann, Fabien Rondepierre.

**Validation:** Isabelle Jalenques, Andreas Hartmann, Sophie Lauron.

**Visualization:** Diane Cyrille, Philippe Derost.

**Writing – original draft:** Isabelle Jalenques, Candy Guiguet-Auclair, Fabien Rondepierre.

**Writing – review & editing:** Andreas Hartmann, Sophie Lauron, Urbain Tauveron—Jalenques.

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
