## [Decision Letter · Decision Letter 0]

4 Jul 2022

PONE-D-22-04149The French version of the Gilles de la Tourette Syndrome Quality of Life Scale for adolescents (GTS-QOL-French-Ado): adaptation and psychometric evaluationPLOS ONE

Dear Dr. Jalenques,

Thank you for submitting your manuscript to PLOS ONE. After careful consideration, we feel that it has merit but does not fully meet PLOS ONE’s publication criteria as it currently stands. Therefore, we invite you to submit a revised version of the manuscript that addresses the points raised during the review process.

The manuscript has been evaluated by two reviewers, and their comments are available below.

The reviewers have raised a number of major concerns. They request improvements to the reporting of methodological aspects of the study such  as the inclusion of information on the sample size and the participants’ characteristics.

Could you please carefully revise the manuscript to address all comments raised?

We look forward to receiving your revised manuscript.

Kind regards,

Lorena Verduci

Staff Editor

PLOS ONE

Journal Requirements:

3. One of the noted authors is a group or consortium for The Syndrome de Gilles de La Tourette Study Group. In addition to naming the author group, please list the individual authors and affiliations within this group in the acknowledgments section of your manuscript. Please also indicate clearly a lead author for this group along with a contact email address.

Additional Editor Comments:

PLOS ONE has specific requirements for studies that are presenting a new method or tool as the primary focus, including a newly developed or modified questionnaire or scale (https://journals.plos.org/plosone/s/submission-guidelines#loc-methods-software-databases-and-tools.) One requirement is that the questionnaire or scale must be openly available under a license no more restrictive than CC BY. In light of this, please include a copy of your questionnaire or scale as a Supporting Information file or provide a link if it is available through an online repository. Also, in your Methods section, please discuss whether you obtained the necessary permissions from the owner of the original questionnaire to modify it.

Reviewers' comments:

Reviewer's Responses to Questions

**Comments to the Author**

1. Is the manuscript technically sound, and do the data support the conclusions?

Reviewer #1: Yes

Reviewer #2: Partly

2. Has the statistical analysis been performed appropriately and rigorously? 

Reviewer #1: Yes

Reviewer #2: I Don't Know

3. Have the authors made all data underlying the findings in their manuscript fully available?

Reviewer #1: No

Reviewer #2: Yes

4. Is the manuscript presented in an intelligible fashion and written in standard English?

Reviewer #1: Yes

Reviewer #2: Yes

5. Review Comments to the Author

Reviewer #1: This study assessed the psychometric properties of the ‘French version of the Gilles the la Tourette Syndrome Quality of Life Scale for adolescents’. The authors did a great job in producing a nicely written and succinct paper.

While I see value in conducting this research, I do have a few questions and/or concerns. Please see my comments below. As someone who has been on the receiving end of criticism through the peer-review process, I try to make helpful comments. I hope these comments can be seen as such.

Firstly, I wonder how robust these results are considering that a small sample is used here. I imagine that accessing relevant participants may be difficult; however, this important limitation should be acknowledged/highlighted.

Secondly, the authors suggest that the HRQol for French-speaking adolescents has good psychometric properties. However, the factorial structure reported here does not seem to be in line with previous literature. More importantly, given that several items showed high cross-loadings, I wonder if the authors’ claim should be rephrased acknowledging these limitations.

Thirdly, I was hoping to see a scree plot and a figure showing the factorial structure.

Fourthly, given that several items were cross-loading, I wonder if a CFA would indicate that the factorial structure presented here is indeed appropriate. I think the sample size represents (again) a major inconvenience to conduct a CFA here. I think the lack of GOF via CFA is probably a big hurdle and makes me question the actual contribution this paper makes.

P7 – Data completeness: “Data quality was considered to be satisfactory if less than 5% of the item data were missing”. – Reference needed.

P7 – Factor analysis: “The perception and definition…” – This and the following sentence do not belong in this section. This information should probably be moved to the introduction section.

P19 – Internal consistency: It is suggested that authors include McDonalds’ omega, and partial consistency when dropping items.

Reviewer #2: The study aimed to validate a new measure of Health Related Quality of Life (HRQoL) which is specifically targeted to French adolescents diagnosed with Gilles de la Tourette syndrome (GTS), adapting their previously published scale for French adults (GTS-QOL-French; Jalenques et al., 2020). The authors have given a very detailed and thorough explanation of their methodology and statistical analyses performed.

There were several statistical shortcomings of the analysis which the authors highlighted and reported in many cases; for example, several items were weighted onto multiple subscales and were subsequently retained or moved to other subscales, these seemed like logical decisions.

In my opinion however, there were several concerns regarding the study:

1. The study mentions the validation of the GTS-QOL for children and adolescents (C&A-GTS-QOL) within the introduction, yet goes on to say that this item was unavailable when the current scale was adapted during phase 1. Phase 1 relates to the same authors’ publication of the GTS-QOL-French (Jalenques et al., 2020) and also references the C&A-GTS-QOL. The C&A-GTS-QOL was validated in Italian in 2013 (Cavanna et al., 2013), and English in 2017 (Su et al., 2017). The authors have referenced all three of these articles in justifying their decision to adapt the GTS-QOL-French-Ado in the current study. I do not understand the authors’ statement that the scale was unavailable. An explanation of this statement and timeline is warranted. The authors could have simply distinguished and justified the need for a HRQoL measure specifically targeted to adolescents which differs from the C&A-GTS-QOL. Due to typically early diagnosis of GTS, together with stages of life and development specific to adolescence, I’m sure the need for a scale suited specifically to adolescents could be justified, however the authors fail to address this.

2. The authors reference Tarwee et al. (2007) for determining the sample size for the study; however this report highlights that as a rule of thumb, 4-10 participants are required per variable with a minimum of 100 participants. A minimum of 50 participants should be considered for measures of reliability. The authors explicitly reported these requirements in their previous publication (Jalenques et al., 2020). The current study included a total of 84 participants with 26 respondents included for test-retest analysis. Tarwee et al. (2007) also suggest that where a theoretical model exists, confirmatory factor analysis should be used. As this study modelled the adaptation of a pre-existing scale for adults to an adolescent population, and given the authors confidence in the results of the EFA, use of CFA could be justified and would provide further credit to the design of the scale. It could be that CFA was not conducted due to the sample size.

3. Given the sample size, a significant number of participants (28%) were excluded from the test-retest analysis; examples of “any event that might have impaired their quality of life” are not provided. In relation to change in health status or recent life events, did the authors screen for such events prior to the initial evaluation? What was the threshold for exclusion from the reliability analysis?

4. In measuring internal consistency, ‘Cognitive’ and ‘Physical’ subscales did not meet the minimum Cronbach’s α coefficient of .70. While the ‘Cognitive’ subscale was close at .68, the ‘Physical’ subscale was only reported at .56.

5. The authors note an addition of the social subscale compared to other studies, which seems novel and highly relevant considering the assessment of quality of life. This subscale however, did not stand up in the test-retest reliability analysis. This could simply be due to sample size in the reliability analysis, or could indicate shortcomings in the results of the EFA, misinterpretation of question/s or interpretation of subscale headings (I note that ‘depressed mood weighed far higher on the ‘social’ subscale than on the ‘psychological’ subscale).

6. There were a large percentage (49.4%) of participants who received help in completing the questionnaire as highlighted by the authors. The authors also noted that there was no statistical difference between those who received help and those who did not in relation to age and health status. While correlations exist between self-report and parent-report of QOL for younger children, Storch et al. (2007) found no correlation in relation to adolescents. Were there statistical differences in relation to QOL between those who received help and those who did not, indicative of potential parental bias?

7. I found it confusing to find and understand the participant data as this has been described over several different sections of the report - Number of participants described in phase 1 of Methods section; Age range reported under participant subheading in Method without further breakdown; Descriptives provided under participant subheading in Results; participants selected for retest described under Reliability subheading in Results. It would be helpful for the reader if participant information was reported concisely in one section.

The authors have clearly worked hard to report a rigorous and complex analysis of the data, justifying their analytic processes and reporting the shortcomings within the results. However, I feel it is difficult to validate the results of the current study in line with the authors’ view, particularly given the sample sizes used and my interpretation of the reported results.

References:

Cavanna, A. E., Luoni, C., Selvini, C., Blangiardo, R., Eddy, C. M., Silvestri, P. R., Cali, P. V., Seri. S., Balottin, U., Cardona, F., Rizzo, R., & Termine, C. (2013). The Gilles De La Tourette Syndrome-Quality of Life Scale for Children and Adolescents (C&A-GTS-QOL): Development and Validation of the Italian Version. Behavioural Neurology, 27(1), 95–103.

Jalenques, I., Cyrille, D., Derost, P., Hartmann, A., Lauron, S., Jameux, C., Tauveron-Jalenques, U., Guiguet-Auclair, C., Rondepierre, F., & Syndrome de Gilles de La Tourette Study Group (2020). Cross-cultural adaptation and psychometric evaluation of the French version of the Gilles de la Tourette Syndrome Quality of Life Scale (GTS-QOL). PloS one, 15(12), e0243912. https://doi.org/10.1371/journal.pone.0243912

Storch, E., Merlo, L., Lack, C., Milsom, V., Geffken, G., Goodman, W., & Murphy, T. (2007). Quality of Life in Youth With Tourette’s Syndrome and Chronic Tic Disorder. Journal of Clinical Child & Adolescent Psychology, 36(2), 217–227.

Su, M. T., Heyman, I., Murphy, T., McFarlane, F., Murray, I., Heidemeyer, L., Cavanna, A. E., & Termine, C. (2017). The English Version of the Gilles de la Tourette Syndrome-Quality of Life Scale for Children and Adolescents (C&A-GTS-QOL): A Validation Study in the United Kingdom. Journal of Child Neurology, 32(1), 76–83.

6. PLOS authors have the option to publish the peer review history of their article (what does this mean?). If published, this will include your full peer review and any attached files.

Reviewer #1: No

Reviewer #2: No

---

## [Author Response · Author response to Decision Letter 0]

7 Sep 2022

To the editor

Style requirement has been checked and changed when required.

The data are available at Mendeley. This information has already been providing in the method section: “All data are available at Mendeley [29].”

29. Jalenques I, Guiguet-Auclair C, Cyrille D, Debosque C, Derost P, Hartmann A, et al. The French version of the Gilles de la Tourette Syndrome Quality of Life Scale for adolescents (GTS-QOL-French-Ado): adaptation and psychometric evaluation. Mendeley Data; 2022. Report No.: V1. Available from: https://data.mendeley.com/datasets/535ctxxsn5/1

3. One of the noted authors is a group or consortium for The Syndrome de Gilles de La Tourette Study Group. In addition to naming the author group, please list the individual authors and affiliations within this group in the acknowledgments section of your manuscript. Please also indicate clearly a lead author for this group along with a contact email address.

Membership of the The Syndrome de Gilles de La Tourette Study Group is provided in the acknowledgments and the lead leader for this group is clearly indicated.

Additional Editor Comments:

PLOS ONE has specific requirements for studies that are presenting a new method or tool as the primary focus, including a newly developed or modified questionnaire or scale (https://journals.plos.org/plosone/s/submission-guidelines#loc-methods-software-databases-and-tools.) One requirement is that the questionnaire or scale must be openly available under a license no more restrictive than CC BY. In light of this, please include a copy of your questionnaire or scale as a Supporting Information file or provide a link if it is available through an online repository. Also, in your Methods section, please discuss whether you obtained the necessary permissions from the owner of the original questionnaire to modify it.

The questionnaire has been added as a supporting information (S1 File).

 

Reviewer #1: This study assessed the psychometric properties of the ‘French version of the Gilles the la Tourette Syndrome Quality of Life Scale for adolescents’. The authors did a great job in producing a nicely written and succinct paper.

While I see value in conducting this research, I do have a few questions and/or concerns. Please see my comments below. As someone who has been on the receiving end of criticism through the peer-review process, I try to make helpful comments. I hope these comments can be seen as such.

Comment 1: Firstly, I wonder how robust these results are considering that a small sample is used here. I imagine that accessing relevant participants may be difficult; however, this important limitation should be acknowledged/highlighted.

Response 1: Accessing relevant participants to our study is indeed difficult. However, to our knowledge, our study assessing HRQoL is one of those including the higher number of adolescents with GTS 1-7. We added this in the discussion.

We highlighted the sample size limitation in the discussion and conclusion of the manuscript, as in the abstract. 

In the discussion of the manuscript: “The major limitation of our study concerns the sample size. The recommended sample size for the factor analysis and the reliability test could not be included 8-9. We obtained a ratio of 3.1 respondents for each question, which might be a limitation for the factor analysis. The factorial structure of the GTS-QOL-French-Ado showed cross-loadings for 5 items. Neither the English-language nor the Italian C&A-GTS-QOL validations indicated whether items loaded on several subscales 10-11. Nevertheless, in the adapted COSMIN Risk of Bias checklist 9, no recommendation was made on the sample size required for reliability analysis and good agreement was obtained in our study for all subscales despite the limited number of participants. Further evaluation of the psychometric properties of the GTS-QOL-French-Ado are needed on larger samples to confirm our first results, and notably its five-factor structure”.

In the conclusion of the manuscript: “Further studies on larger sample to confirm these first results and longitudinal studies to assess responsiveness to change with therapeutic interventions would be of great interest”.

In the discussion of abstract: “The GTS-QOL-French-Ado is the first disease-specific HRQoL tool for French-speaking adolescents with GTS aged 12-16 years, and shows good psychometric properties. Further psychometric testing is recommended to confirm our first results”.

We modified the sentence at the end of the ‘Study design’ paragraph, to refer to the COSMIN checklist 9: “A sample size of more than 50 subjects is rated good by the COSMIN group for the internal consistency evaluation. A minimum number of 100 subjects or six times the number of items is recommended to ensure satisfactory factor analysis. A sample size of at least 50 subjects is recommended to guarantee acceptable assessment for reliability”. 

Our study included 84 adolescents in the structural analysis and 32 in the reliability analysis. So, our study is rated good for internal consistency but not for factor and test-retest reliability analysis. However, in the adapted COSMIN Risk of Bias checklist 9, no recommendation was made on the sample size required for reliability analysis. Despite the small sample size for the reliability analysis, good agreement was obtained for all the subscales.

1 Evans J, Seri S, Cavanna AE. The effects of Gilles de la Tourette syndrome and other chronic tic disorders on quality of life across the lifespan: a systematic review. Eur Child Adolesc Psychiatry. 2016;25:939‑948.

2 O’Hare D, Helmes E, Reece J, Eapen V, McBain K. The Differential Impact of Tourette’s Syndrome and Comorbid Diagnosis on the Quality of Life and Functioning of Diagnosed Children and Adolescents: The Differential Impact of Tourette’s Syndrome and Comorbid Diagnosis on the Quality of Life and Functioning of Diagnosed Children and Adolescents. J Child Adolesc Psychiatr Nurs. 2016;29:30‑36.

3 Dabrowski J, King J, Edwards K, Yates R, Heyman I, Zimmerman-Brenner S, et al. The Long-Term Effects of Group-Based Psychological Interventions for Children With Tourette Syndrome: A Randomized Controlled Trial. Behavior Therapy. 2018;49:331‑343.

4 Rizzo R, Pellico A, Silvestri P, Chiarotti F, Cardona F. A randomized controlled trial comparing behavioral, educational, and pharmacological treatments in youths with chronic tic disorder or Tourette syndrome. Frontiers in psychiatry. 2018;9. doi/10.3389/fpsyt.2018.00100.

5 Erbilgin Gün S, Kilincaslan A. Quality of Life Among Children and Adolescents With Tourette Disorder and Comorbid ADHD: A Clinical Controlled Study. J Atten Disord. 2019;23:817‑827. 

6 Solís García G, Jové Blanco A, Chacón Pascual A, Vázquez López M, Castro de Castro P, Carballo Belloso JJ, et al. Calidad de vida y comorbilidades psiquiátricas en pacientes pediátricos con síndrome de Gilles de la Tourette. RevNeurol. 2021;73:339. 

7 Coffey B, Jankovic J, Claassen DO, Jimenez-Shahed J, Gertz BJ, Garofalo EA, et al. Efficacy and Safety of Fixed-Dose Deutetrabenazine in Children and Adolescents for Tics Associated With Tourette Syndrome: A Randomized Clinical Trial. JAMA Netw Open. 2021;4:e2129397. 

8 COSMIN - Improving the selection of outcome measurement instruments. COSMIN. Available from: https://www.cosmin.nl/

9 Mokkink LB, de Vet HCW, Prinsen CAC, Patrick DL, Alonso J, Bouter LM, Terwee CB. COSMIN Risk of Bias checklist for systematic reviews of Patient-Reported Outcome Measures. Qual. Life Res. 2018; 27:1171–1179. 

10 Cavanna AE, Chiara L, Claudia S, Rosanna B, Eddy CM, Silvestri PR, et al. The Gilles de la Tourette Syndrome-Quality of Life Scale for children and adolescents (C&A-GTS-QOL): Development and validation of the Italian version. Behavioural Neurology. 2013;(1):95‑103. 

11 Su MT, McFarlane F, Cavanna AE, Termine C, Murray I, Heidemeyer L, et al. The English Version of the Gilles de la Tourette Syndrome-Quality of Life Scale for Children and Adolescents (C&A-GTS-QOL). J Child Neurol. 2017;32:76‑83. 

Comment 2: Secondly, the authors suggest that the HRQol for French-speaking adolescents has good psychometric properties. However, the factorial structure reported here does not seem to be in line with previous literature. More importantly, given that several items showed high cross-loadings, I wonder if the authors’ claim should be rephrased acknowledging these limitations.

Response 2: We modified the first sentence of the discussion: “Our study described the evaluation of the psychometric properties of the GTS-QOL-French-Ado”. 

We discussed the difference between the factorial structure of the GTS-QOL-French-Ado, the GTS-QOL-French for adults and the Italian and English versions of the C&A-GTS-QOL, which were not the same. We reported the cross-loadings found in the factor analysis in our study, which was not the case in the validation of the Italian and English versions of the C&A-GTS-QOL. So, we could not certify that in these versions, there was no cross-loadings of the items. In the validation of the Italian version, it was stated that there was no cross-loadings, but the definition of cross-loading was not reported. 

We added a sentence in the discussion on this limitation: “The factorial structure of the GTS-QOL-French-Ado showed cross-loadings for 5 items. Neither the English-language nor the Italian C&A-GTS-QOL validations indicated whether items loaded on several subscales (…). Further evaluation of the psychometric properties of the GTS-QOL-French-Ado are needed on larger samples to confirm our first results, and notably its five-factor structure”. 

Comment 3: Thirdly, I was hoping to see a scree plot and a figure showing the factorial structure.

Response 3: A scree plot and a figure showing the factorial structure were added (S1 Fig. Scree plot of the eigenvalues for the factor analysis of the GTS-QOL-French-Ado; Fig 1. Path diagram of the factorial structure of the GTS-QOL-French-Ado from the exploratory factor analysis). 

Comment 4: Fourthly, given that several items were cross-loading, I wonder if a CFA would indicate that the factorial structure presented here is indeed appropriate. I think the sample size represents (again) a major inconvenience to conduct a CFA here. I think the lack of GOF via CFA is probably a big hurdle and makes me question the actual contribution this paper makes.

Response 4: The sample size of our study is indeed a major inconvenience to conduct a CFA. Anthoine et al. 1 reported that recommendations for the sample size when conducting a CFA are disparate, ranging from 150 to 1000 subjects. The large sample size is a more crucial requirement in CFA compared to EFA 2.

Moreover, we chose to conduct an EFA in reason of the different factor structure found for the GTS-QOL or the C&A-GTS-QOL. Our French version of the GTS-QOL for adults showed a different factor structure in comparison with the English original version. For adolescents, two different factor structures were found in the Italian and the English validation of the C&A-GTS-QOL. We used EFA to explore the possible underlying factor structure of a set of observed variables without imposing a preconceived structure on the outcome. 

In the framework of a CFA, a hypothesized model is established and the CFA confirm that it provides a good fit to the observed data. This makes CFA a method suited for validation of instruments with a predefined measurement model. Or, given previous studies on the GTS-QOL for adults or adapted for adolescents that found different factor structures, such hypothesized model was lacking. 

1 Anthoine E, Moret L, Regnault A, Sbille V, Hardouin JB. Sample size used to validate a scale: a review of publications on newly-developed patient reported outcomes measures. Health and Quality of Life Outcomes. 2014;12:176.

2 Kyriazos TA. Applied Psychometrics: Sample Size and Sample Power Considerations in Factor Analysis (EFA, CFA) and SEM in General. Psychology. 2018; 9:2207-2230. 

Comment 5: P7 – Data completeness: “Data quality was considered to be satisfactory if less than 5% of the item data were missing”. – Reference needed.

Response 5: We deleted this sentence. To our knowledge, no existing literature refer to a threshold that allow to consider the percentage of missing values as satisfactory. Moreover, Mokkink et al. 1 decided to remove standards on missing data in the adaptation of the original COSMIN checklist. They consider “information on missing items very important to report”1 but decided to remove these standards.

1 Mokkink, L.B.; de Vet, H.C.W. ; Prinsen, C.A.C. ; Patrick, D.L. ; Alonso, J. ; Bouter, L.M.; Terwee, C.B.. COSMIN Risk of Bias checklist for systematic reviews of Patient-Reported Outcome Measures. Qual. Life Res. 2018, 27, 1171–1179. DOI: 10.1007/s11136-017-1765-4.

Comment 6: P7 – Factor analysis: “The perception and definition…” – This and the following sentence do not belong in this section. This information should probably be moved to the introduction section.

Response 6: The sentence was moved in the last paragraph of the introduction and modified: “The perception and definition of HRQoL may vary from one population to another or by generation, and so there was no guarantee that the French version of the GTS-QOL for adolescents faithfully reflected the subscales of the questionnaire for adults. We therefore decided to perform a full-scale psychometric assessment and to explore the factorial structure of the GTS-QOL for adolescents with an exploratory factor analysis of the items.”

Comment 7: P19 – Internal consistency: It is suggested that authors include McDonalds’ omega, and partial consistency when dropping items.

Response 7: We included McDonalds’ omega to better assess the internal consistency of the GTS-QOL-French-Ado subscales in the corresponding subheading of the Statistical analysis and Results sections. Corrected item-total correlations are reported in S2 Table and described in the Item-total correlations subheading of the Results section.

 

Reviewer #2: The study aimed to validate a new measure of Health Related Quality of Life (HRQoL) which is specifically targeted to French adolescents diagnosed with Gilles de la Tourette syndrome (GTS), adapting their previously published scale for French adults (GTS-QOL-French; Jalenques et al., 2020). The authors have given a very detailed and thorough explanation of their methodology and statistical analyses performed.

There were several statistical shortcomings of the analysis which the authors highlighted and reported in many cases; for example, several items were weighted onto multiple subscales and were subsequently retained or moved to other subscales, these seemed like logical decisions.

In my opinion however, there were several concerns regarding the study:

Comment 1: The study mentions the validation of the GTS-QOL for children and adolescents (C&A-GTS-QOL) within the introduction, yet goes on to say that this item was unavailable when the current scale was adapted during phase 1. Phase 1 relates to the same authors’ publication of the GTS-QOL-French (Jalenques et al., 2020) and also references the C&A-GTS-QOL. The C&A-GTS-QOL was validated in Italian in 2013 (Cavanna et al., 2013), and English in 2017 (Su et al., 2017). The authors have referenced all three of these articles in justifying their decision to adapt the GTS-QOL-French-Ado in the current study. I do not understand the authors’ statement that the scale was unavailable. An explanation of this statement and timeline is warranted. The authors could have simply distinguished and justified the need for a HRQoL measure specifically targeted to adolescents which differs from the C&A-GTS-QOL. Due to typically early diagnosis of GTS, together with stages of life and development specific to adolescence, I’m sure the need for a scale suited specifically to adolescents could be justified, however the authors fail to address this.

Response 1: The introduction was modified to take into account this comment. We now justify the need for a HRQoL measure specifically targeted to adolescents, as no such instrument was available in French. Our team had validated a French version of the GTS-QOL, a self-rated disease-specific questionnaire for adults with GTS. Thus, we decided to adapt this specific instrument to adolescents aged between 12 and 16 years old.

In the introduction of the manuscript: “Our team had validated a French version of the GTS-QOL for adults 1 that we decided to adapt for French speaking adolescents. Indeed, to our knowledge, there has been only one study in France that has assessed the HRQoL of adolescents with GTS, using the generic HRQoL instrument “ Vécu et Santé Perçue de l’Adolescent” (VSP-A) 2. Nevertheless, the early diagnosis of GTS, the stages of life and development specific to adolescence, necessitate a scale suited specifically to adolescents, especially since studies have highlighted differences in HRQoL domains across different age groups of patients with GTS 3,4”.

We also deleted the reference to the validation of the C&A-GTS-QOL in Italian and in English in the introduction to avoid confusion. These two versions of the C&A-GTS-QOL were compared to the GTS-QOL-French-Ado in the discussion. We explained in the discussion why these versions of the C&A-GTS-QOL were unavailable : “In parallel of our work (that began in 2012), two Italian and English versions of the GTS-QOL for children and adolescents, named the C&A-GTS-QOL, were developed and validated in 2013 and 2016 ”.

1 Jalenques I, Cyrille D, Derost P, Hartmann A, Lauron S, Jameux C, et al. Cross-cultural adaptation and psychometric evaluation of the French version of the Gilles de la Tourette Syndrome Quality of Life Scale (GTS-QOL). Carvalho S, éditeur. PLoS ONE. 2020;15:e0243912. 

2 Jalenques I, Auclair C, Morand D, Legrand G, Marcheix M, Ramanoel C, et al. Health-related quality of life, anxiety and depression in parents of adolescents with Gilles de la Tourette syndrome: a controlled study. Eur Child Adolesc Psychiatry. 2017;26:603‑617. 

3 Evans J, Seri S, Cavanna AE. The effects of Gilles de la Tourette syndrome and other chronic tic disorders on quality of life across the lifespan: a systematic review. Eur Child Adolesc Psychiatry. 2016;25:939‑948.

4 Silvestri PR, Chiarotti F, Baglioni V, Neri V, Cardona F, Cavanna AE. Health-related quality of life in patients with Gilles de la Tourette syndrome at the transition between adolescence and adulthood. Neurol Sci. 2016;37:1857‑1860. 

Comment 2: The authors reference Tarwee et al. (2007) for determining the sample size for the study; however this report highlights that as a rule of thumb, 4-10 participants are required per variable with a minimum of 100 participants. A minimum of 50 participants should be considered for measures of reliability. The authors explicitly reported these requirements in their previous publication (Jalenques et al., 2020). The current study included a total of 84 participants with 26 respondents included for test-retest analysis. Tarwee et al. (2007) also suggest that where a theoretical model exists, confirmatory factor analysis should be used. As this study modelled the adaptation of a pre-existing scale for adults to an adolescent population, and given the authors confidence in the results of the EFA, use of CFA could be justified and would provide further credit to the design of the scale. It could be that CFA was not conducted due to the sample size.

Response 2: We modified the sentence at the end of the ‘Study design’ paragraph, to refer to the COSMIN checklist 1: “A sample size of more than 50 subjects is rated good by the COSMIN group for the internal consistency evaluation. A minimum number of 100 subjects or six times the number of items is recommended to ensure satisfactory factor analysis. A sample size of at least 50 subjects is recommended to guarantee acceptable assessment for reliability”. 

Our study included 84 adolescents in the structural analysis and 32 in the reliability analysis. So, our study is rated good for internal consistency but not for factor and test-retest reliability analysis. However, in the adapted COSMIN Risk of Bias checklist 1, no recommendation was made on the sample size required for reliability analysis. Despite the small sample size for the reliability analysis, good agreement was obtained for all the subscales.

We highlighted the sample size limitation in the discussion and conclusion of the manuscript, as in the abstract. 

In the discussion of the manuscript: “The major limitation of our study concerns the sample size. The recommended sample size for the factor analysis and the reliability test could not be included 1,2. We obtained a ratio of 3.1 respondents for each question, which might be a limitation for the factor analysis. The factorial structure of the GTS-QOL-French-Ado showed cross-loadings for 5 items. Neither the English-language nor the Italian C&A-GTS-QOL validations indicated whether items loaded on several subscales 3,4. Nevertheless, in the adapted COSMIN Risk of Bias checklist 1, no recommendation was made on the sample size required for reliability analysis and good agreement was obtained in our study for all subscales despite the limited number of participants. Further evaluation of the psychometric properties of the GTS-QOL-French-Ado are needed on larger samples to confirm our first results, and notably its five-factor structure”.

In the conclusion of the manuscript: “Further studies on larger sample to confirm these first results and longitudinal studies to assess responsiveness to change with therapeutic interventions would be of great interest”.

In the discussion of abstract: “The GTS-QOL-French-Ado is the first disease-specific HRQoL tool for French-speaking adolescents with GTS aged 12-16 years, and shows good psychometric properties. Further psychometric testing is recommended to confirm our first results”.

The sample size of our study is a major inconvenience to conduct a CFA. Anthoine et al. 5 reported that recommendations for the sample size when conducting a CFA are disparate, ranging from 150 to 1000 subjects. The large sample size is a more crucial requirement in CFA compared to EFA 6. Moreover, we chose to conduct an EFA in reason of the different factor structure found for the GTS-QOL or the C&A-GTS-QOL. Our French version of the GTS-QOL for adults showed a different factor structure in comparison with the English original version. For adolescents, two different factor structures were found in the Italian and the English validation of the C&A-GTS-QOL. We used EFA to explore the possible underlying factor structure of a set of observed variables without imposing a preconceived structure on the outcome. 

In the framework of a CFA, a hypothesized model is established and the CFA confirm that it provides a good fit to the observed data. This makes CFA a method suited for validation of instruments with a predefined measurement model. Or, given previous studies on the GTS-QOL for adults or adapted for adolescents that found different factor structures, such hypothesized model was lacking. 

1 Mokkink LB, de Vet HCW, Prinsen CAC, Patrick DL, Alonso J, Bouter LM, Terwee CB. COSMIN Risk of Bias checklist for systematic reviews of Patient-Reported Outcome Measures. Qual. Life Res. 2018; 27:1171–1179. 

2 COSMIN - Improving the selection of outcome measurement instruments. COSMIN. Available from: https://www.cosmin.nl/

3 Cavanna AE, Chiara L, Claudia S, Rosanna B, Eddy CM, Silvestri PR, et al. The Gilles de la Tourette Syndrome-Quality of Life Scale for children and adolescents (C&A-GTS-QOL): Development and validation of the Italian version. Behavioural Neurology. 2013;(1):95‑103. 

4 Su MT, McFarlane F, Cavanna AE, Termine C, Murray I, Heidemeyer L, et al. The English Version of the Gilles de la Tourette Syndrome-Quality of Life Scale for Children and Adolescents (C&A-GTS-QOL). J Child Neurol. 2017;32:76‑83. 

5 Anthoine E, Moret L, Regnault A, Sbille V, Hardouin JB. Sample size used to validate a scale: a review of publications on newly-developed patient reported outcomes measures. Health and Quality of Life Outcomes. 2014;12:176.

6 Kyriazos TA. Applied Psychometrics: Sample Size and Sample Power Considerations in Factor Analysis (EFA, CFA) and SEM in General. Psychology. 2018; 9:2207-2230. 

Comment 3: Given the sample size, a significant number of participants (28%) were excluded from the test-retest analysis; examples of “any event that might have impaired their quality of life” are not provided. In relation to change in health status or recent life events, did the authors screen for such events prior to the initial evaluation? What was the threshold for exclusion from the reliability analysis?

Response 3: We thanks the reviewer for this helpful comment. 

We did not screen for change in health status or recent life events prior to the initial evaluation. 

For test-retest reliability, it is important to check any change in health status as recommended in the adapted COSMIN Risk of Bias checklist1. A study is rated “excellent” if evidence is provide to respond yes to the criteria “Were patients stable in the interim period on the construct to be measured”. So we screen about any change in the health status of participants between test ad retest, by asking them if their health status had degraded. If they respond yes, they were excluded for reliability analysis. To take into account the comments of the reviewer, we only excluded participants with degraded health status from reliability analysis. Participants who reported modifications in their treatment since the first evaluation or who described any event that might have impaired their quality of life between test and retest were kept in the analysis, to increase the sample size. So, 32 participants were retained for the reliability analysis.

Results of the reliability analysis and the study design of Phase 2 were modified in consequence (“Respondents who reported any degradation in their health status between test and retest were excluded from the reliability analysis”).

1 Mokkink LB, de Vet HCW, Prinsen CAC, Patrick DL, Alonso J, Bouter LM, Terwee CB. COSMIN Risk of Bias checklist for systematic reviews of Patient-Reported Outcome Measures. Qual. Life Res. 2018; 27:1171–1179. 

Comment 4: In measuring internal consistency, ‘Cognitive’ and ‘Physical’ subscales did not meet the minimum Cronbach’s α coefficient of .70. While the ‘Cognitive’ subscale was close at .68, the ‘Physical’ subscale was only reported at .56.

We highlighted this limitation in the discussion: “The GTS-QOL-French-Ado subscales showed good internal consistency, except for the ‘Physical’ subscale. In the English version of the C&A-GTS-QOL, this dimension showed also moderate internal consistency with a Cronbach’ alpha coefficient value of 0.50”. 

Comment 5: The authors note an addition of the social subscale compared to other studies, which seems novel and highly relevant considering the assessment of quality of life. This subscale however, did not stand up in the test-retest reliability analysis. This could simply be due to sample size in the reliability analysis, or could indicate shortcomings in the results of the EFA, misinterpretation of question/s or interpretation of subscale headings (I note that ‘depressed mood weighed far higher on the ‘social’ subscale than on the ‘psychological’ subscale).

Response 5: We agreed with the reviewer: item 16 “depressed mood” had a loading higher on the ‘Social’ subscale than on the ‘Psychological’ subscale. In the Italian and English versions, this item loaded on a psychological factor that comprised items of the GTS-QOL-French-Ado ‘Social’ subscale like “loneliness/isolation”, “lack of self-confidence”, “lack of social support”, “difficulty seeing friends” in the Italian version and “loneliness/isolation”, “difficulty seeing friends” in the English version. 

Associations of loneliness and social isolation with depressive symptoms among adolescents was recently found by Christiansen et al.1and Viduani et al.2. 

So we decided to place this item in the ‘Social’ subscale and analysis were done again for the modified ‘Psychological’ (without item 16) and ‘Social’ (with item 16) subscales. The results are presented in the revised version of our manuscript. Results were very similar in terms of scores distribution, internal consistency, item-total consistency, inter-subscale correlations and convergent validity. For the reliability analysis, with a sample of 32 participants, ICCs values were higher than 0.70 for the ‘Psychological’ and ‘Social’ subscales. For the ‘Social’ subscale, ICC value raise from 0.61 to 0.74, showing the good reliability of the modified ‘Social’ subscale. 

We modified the discussion: “Item 16 (Depressed mood) overlaps between the ‘Psychological’ and ‘Social’ subscales in the GTS-QOL-French-Ado but we chose to place it in the ‘Social’ subscale, enhancing the clinical relevance of the subscale. Our choice is consistent with recent results concerning associations of loneliness and social isolation with depressive symptoms among adolescents 1,2. Moreover, in the Italian and English versions of the C&A-GTS-QOL, item 16 loaded on a psychological factor that comprised items of the GTS-QOL-French-Ado ‘Social’ subscale”.

1 Christiansen J, Qualter P, Friis K, Pedersen SS, Lund R, Andersen CM, Bekker-Jeppesen M, Lasgaard M. Associations of loneliness and social isolation with physical and mental health among adolescents and young adults. Perspect. Public Health. 2021; 141:226-236. 

2 Viduani A, Benetti S, Martini T, Buchweitz C, Ottman K, Wahid SS, Fisher HL, Mondelli V, Kohrt BA, Kielong C. Social isolation as a core feature of adolescent depression: a qualitative study in Porto Alegre, Brazil. Int. J. Qual. Stud. Health Well-being. 2021; 16:1978374. 

Comment 6: There were a large percentage (49.4%) of participants who received help in completing the questionnaire as highlighted by the authors. The authors also noted that there was no statistical difference between those who received help and those who did not in relation to age and health status. While correlations exist between self-report and parent-report of QOL for younger children, Storch et al. (2007) found no correlation in relation to adolescents. Were there statistical differences in relation to QOL between those who received help and those who did not, indicative of potential parental bias?

Response 6: A difference in relation to QOL between those who received help and those who did not was found for the ‘Cognitive’ subscale with a mean score of 47.7 (SD 22.6) for adolescents who received help versus 34.2 (SD 24.1) for the others. Regarding items of the this subscale, differences were found for item item 11 “difficulty concentrating” (with a mean score of 2.8 (SD 1.3) for adolescents who received help versus 2.3 (SD 1.2) for the others) and item 14 “difficulty finishing tasks” (with a mean score of 2.1 (SD 1.3) for adolescents who received help versus 1.3 (SD 1.4) for the others). 

These results highlighted that adolescents who received help to fill in the questionnaire were more impaired in the ‘Cognitive’ domain, and especially had more difficulty concentrating and more difficulty finishing tasks. As our aim was to assess the psychometric properties of the GTS-QOL-French-Ado, it does not represent a bias in our study. 

Comment 7: I found it confusing to find and understand the participant data as this has been described over several different sections of the report - Number of participants described in phase 1 of Methods section; Age range reported under participant subheading in Method without further breakdown; Descriptives provided under participant subheading in Results; participants selected for retest described under Reliability subheading in Results. It would be helpful for the reader if participant information was reported concisely in one section.

Response 7: As recommended, the participant information was reported in one section, under Participants subheading in Results section. The Number of participants described in Phase 1 of Methods section, the age range reported under Participants subheading in phase 2 of Methods section, and the description of participants selected for retest under Reliability subheading in Results were removed. 

The authors have clearly worked hard to report a rigorous and complex analysis of the data, justifying their analytic processes and reporting the shortcomings within the results. However, I feel it is difficult to validate the results of the current study in line with the authors’ view, particularly given the sample sizes used and my interpretation of the reported results.

References:

Cavanna, A. E., Luoni, C., Selvini, C., Blangiardo, R., Eddy, C. M., Silvestri, P. R., Cali, P. V., Seri. S., Balottin, U., Cardona, F., Rizzo, R., & Termine, C. (2013). The Gilles De La Tourette Syndrome-Quality of Life Scale for Children and Adolescents (C&A-GTS-QOL): Development and Validation of the Italian Version. Behavioural Neurology, 27(1), 95–103.

Jalenques, I., Cyrille, D., Derost, P., Hartmann, A., Lauron, S., Jameux, C., Tauveron-Jalenques, U., Guiguet-Auclair, C., Rondepierre, F., & Syndrome de Gilles de La Tourette Study Group (2020). Cross-cultural adaptation and psychometric evaluation of the French version of the Gilles de la Tourette Syndrome Quality of Life Scale (GTS-QOL). PloS one, 15(12), e0243912. https://doi.org/10.1371/journal.pone.0243912

Storch, E., Merlo, L., Lack, C., Milsom, V., Geffken, G., Goodman, W., & Murphy, T. (2007). Quality of Life in Youth With Tourette’s Syndrome and Chronic Tic Disorder. Journal of Clinical Child & Adolescent Psychology, 36(2), 217–227.

Su, M. T., Heyman, I., Murphy, T., McFarlane, F., Murray, I., Heidemeyer, L., Cavanna, A. E., & Termine, C. (2017). The English Version of the Gilles de la Tourette Syndrome-Quality of Life Scale for Children and Adolescents (C&A-GTS-QOL): A Validation Study in the United Kingdom. Journal of Child Neurology, 32(1), 76–83.

---

## [Decision Letter · Decision Letter 1]

19 Sep 2022

PONE-D-22-04149R1

The French version of the Gilles de la Tourette Syndrome Quality of Life Scale for adolescents (GTS-QOL-French-Ado): adaptation and psychometric evaluation

PLOS ONE

Dear Dr. Jalenques,

Thank you for submitting your manuscript to PLOS ONE. After careful consideration, we have decided that your manuscript does not meet our criteria for publication and must therefore be rejected.

Specifically

**Plagiarism is not acceptable in PLOS submissions**. Plagiarized content will not be considered for publication.

I am sorry that we cannot be more positive on this occasion, but hope that you appreciate the reasons for this decision.

Kind regards,

Nabi Nazari, Phd

Academic Editor

PLOS ONE

Additional Editor Comments:

Authors should be aware that replication of text from their own previous publications is text recycling (also referred to as self-plagiarism), and in some cases is considered unacceptable ( see attachment).

Reviewers' comments:

Reviewer's Responses to Questions

**Comments to the Author**

1. If the authors have adequately addressed your comments raised in a previous round of review and you feel that this manuscript is now acceptable for publication, you may indicate that here to bypass the “Comments to the Author” section, enter your conflict of interest statement in the “Confidential to Editor” section, and submit your "Accept" recommendation.

Reviewer #1: All comments have been addressed

2. Is the manuscript technically sound, and do the data support the conclusions?

Reviewer #1: Yes

3. Has the statistical analysis been performed appropriately and rigorously? 

Reviewer #1: Yes

4. Have the authors made all data underlying the findings in their manuscript fully available?

Reviewer #1: Yes

5. Is the manuscript presented in an intelligible fashion and written in standard English?

Reviewer #1: Yes

6. Review Comments to the Author

Reviewer #1: Thank you for the opportunity to review the revised version of the manuscript titled ‘French version of the Gilles the la Tourette Syndrome Quality of Life Scale for adolescents’.

This version represents a significant improvement where all my previous comments have been addressed.

However, I do have a new minor comment/suggestion. Specifically, in the statistical analysis section, the authors state "statistical analyses were […] conducted at a two-sided alpha 0.05 sig level”. It would good be clarify which analyses is this statement referring to.

I wish the authors good luck with their future publications.

7. PLOS authors have the option to publish the peer review history of their article (what does this mean?). If published, this will include your full peer review and any attached files.

Reviewer #1: No

- - - - -

---

## [Author Response · Author response to Decision Letter 1]

17 Oct 2022

Reviewer #1: Thank you for the opportunity to review the revised version of the manuscript titled ‘French version of the Gilles the la Tourette Syndrome Quality of Life Scale for adolescents’.

This version represents a significant improvement where all my previous comments have been addressed.

However, I do have a new minor comment/suggestion. Specifically, in the statistical analysis section, the authors state "statistical analyses were […] conducted at a two-sided alpha 0.05 sig level”. It would good be clarify which analyses is this statement referring to.

I wish the authors good luck with their future publications.

We thank the reviewer whose suggestions helped to improve the article.

We have specified the analysis to which the significance level referred to in the statistical analysis section: “Statistical analyses were performed with SAS v9.4 software (SAS Institute) and conducted at a two-sided alpha = 0.05 significance level to test the null hypothesis of zero correlation between two variables”.

---

## [Editor Report · Decision Letter 2]

2 Nov 2022

PONE-D-22-04149R2

The French version of the Gilles de la Tourette Syndrome Quality of Life Scale for adolescents (GTS-QOL-French-Ado): adaptation and psychometric evaluation

PLOS ONE

Dear Dr. Jalenques,

Thank you for submitting your manuscript to PLOS ONE. After careful consideration, we feel that it has merit but does not fully meet PLOS ONE’s publication criteria as it currently stands. Therefore, we invite you to submit a revised version of the manuscript that addresses the points raised during the review process.

We look forward to receiving your revised manuscript.

Kind regards,

Charlotte Beaudart

Academic Editor

PLOS ONE

Journal Requirements:

Additional Editor Comments (if provided):

Dear Authors,

I had the opportunity to consider your appeal request and revise your manuscript.

I found your manuscript very clear and detailed. However, I have some comments that I would like you to consider before I could make my decision about the publication of this work.

Please find my comments below:

Please report the 95%CI of ICC in the abstract.

In the conclusion of the abstract, please be a little more specific with the sentence : “Further psychometric testing is recommended to confirm our first results”. Indeed, it is surprising to read this sentence just after giving results of psychometric testing in your own study. Which psychometric properties should be tested again? Or why this should be tested again? Please details.

In page 4, phase 1, please adapt the sentence “Our team cross-culturally adapted the GTS-QOL in adults (GTS-QOL-French)” with “Our team translated in French and cross-culturally adapted the GTS-QOL in adults (GTS-QOL-French)”. It will make the sentence more clear.

It is not clear what was the time interval used for test-retest reliability. In the method section, authors mentioned 2 weeks. However, in the results section, authors mentioned 1 month. Moreover, it is not clear why only a sample of respondents were invited to complete a second time the questionnaire. The sample size for test-restest reliability is therefore too small to obtain valid results.

In the results section, please merge the two paragraph about (i.e. the paragraph presenting the number of participants included in this analysis and the paragraph presenting results of reliability).

Please report the percentage of hypotheses confirmed for convergent validity. It is suggested that a convergent validity is ensured if at least 75% of hypotheses are met.

Are the French versions of the CDI and MASC validated ? I asked the question because authors provided results of psychometric analyses of the French version of the MOVES but for the two other scales used.

Regarding sociodemographic data, I absolutely do not understand the sentence “All data are available at Mendeley [29]”.

Regarding factorial analysis, it is not clear if previous work already proposed a theoretical model for the original scale. If yes, a CFA should be performed instead of a EFA (see one previous comment of reviewer for which authors did not replied “Terwee et al. (2007) also suggest that where a theoretical model exists, confirmatory factor analysis should be used. As this study modelled the adaptation of a pre-existing scale for adults to an adolescent population, and given the authors confidence in the results of the EFA, use of CFA could be justified and would provide further credit to the design of the scale”. I have the same comment, I do not understand why authors did not performed a CFA since a theoretical model already exists”.
---

## [Author Response · Author response to Decision Letter 2]

7 Nov 2022

Journal Requirements

Response: All the references have been checked: no cited papers have been retracted. Two changes have been done in the reference list: the journal names of the references 17 and 21 have been modified.

To the editor

Comment 1: Please report the 95%CI of ICC in the abstract.

Response 1: As recommended, we reported in the abstract the 95% CI of ICC. 

Comment 2: In the conclusion of the abstract, please be a little more specific with the sentence: “Further psychometric testing is recommended to confirm our first results”. Indeed, it is surprising to read this sentence just after giving results of psychometric testing in your own study. Which psychometric properties should be tested again? Or why this should be tested again? Please details.

Response 2: We modified the sentence to highlight that further studies on responsiveness to change would be of great interest. 

Comment 3: In page 4, phase 1, please adapt the sentence “Our team cross-culturally adapted the GTS-QOL in adults (GTS-QOL-French)” with “Our team translated in French and cross-culturally adapted the GTS-QOL in adults (GTS-QOL-French)”. It will make the sentence more clear.

Response 3: We modified the sentence as suggested. 

Comment 4: It is not clear what was the time interval used for test-retest reliability. In the method section, authors mentioned 2 weeks. However, in the results section, authors mentioned 1 month. 

Response 4: For test-retest reliability, the GTS-QOL-French-Ado questionnaire was mailed a second time 15 days after the first assessment. Then the adolescents returned by mail their completed questionnaire. We considered for test-retest reliability analysis only the questionnaires received by mail within 15 days, which corresponds to one month after the first completion. We agree that this was not clear and we modified the sentence in the results section: “For the reliability assessment, the retest was sent to 46 participants. Of the 36 (78.3%) who returned completed questionnaires within 15 days, four reported that their health status had degraded between test and retest”. 

Comment 5: Moreover, it is not clear why only a sample of respondents were invited to complete a second time the questionnaire. The sample size for test-restest reliability is therefore too small to obtain valid results. 

Response 5: We first invited a sample of respondents to complete a second time the GTS-QOL-French-Ado questionnaire. The response rate was very low and we decided in consequence to invite all adolescents subsequently included to complete a second time the questionnaire. Unfortunately, the resulted number of participants in the test-retest reliability was lower than the one attempted, that is 50 participants. 

However, as highlighted in the discussion section, in the adapted COSMIN Risk of Bias checklist 1, no recommendation was made on the sample size required for reliability analysis. Despite the small sample size for the reliability analysis, good agreement was obtained for all the subscales. 

1 Mokkink LB, de Vet HCW, Prinsen CAC, Patrick DL, Alonso J, Bouter LM, Terwee CB. COSMIN Risk of Bias checklist for systematic reviews of Patient-Reported Outcome Measures. Qual. Life Res. 2018; 27:1171–1179. 

Comment 6: In the results section, please merge the two paragraph about (i.e. the paragraph presenting the number of participants included in this analysis and the paragraph presenting results of reliability). 

Response 6: A reviewer asked us previously to report concisely in one section all the participants’ data. As suggested here, we merged the paragraph presenting the number of participants included in the reliability analysis and the paragraph presenting the results of reliability under Reliability subheading in Results section. 

Comment 7: Please report the percentage of hypotheses confirmed for convergent validity. It is suggested that a convergent validity is ensured if at least 75% of hypotheses are met. 

Response 7: As suggested, we reported the percentage of hypotheses confirmed for convergent validity under Convergent validity subheading in Results section. We also added in the Discussion section that 92.2% of hypotheses were met. 

Comment 8: Are the French versions of the CDI and MASC validated ? I asked the question because authors provided results of psychometric analyses of the French version of the MOVES but for the two other scales used. 

Response 8: The French versions of the CDI and MASC have been validated by Mack et al. 1 and Turgeon et al. 2 respectively. We agree with the editor that it was not clear as we reported results of psychometric analysis of the French version of the MOVES. We deleted this sentence. 

1 Mack K, Moor L. Versions françaises d’échelles d’évaluation de la dépression: Les échelles CDI et ISC de Maria Kovacs. 1982;30:627‑652.

2 Turgeon L, Chartrand É, Robaey P, Gauthier AK. [Psychometric properties of a French version of the Multidimensional Anxiety Scale for Children (MASC).]. Revue Francophone de Clinique Comportementale et Cognitive. 2006;11. 

Comment 9: Regarding sociodemographic data, I absolutely do not understand the sentence “All data are available at Mendeley [29]”. 

Response 9: We provided here the public repository where the data underlying the findings presented in the manuscript are held. It is a data availability statement describing where the data can be found. If it is not clear, this sentence could be deleted, as the data availability statement could be found elsewhere in PLoS ONE publication.

Comment 10: Regarding factorial analysis, it is not clear if previous work already proposed a theoretical model for the original scale. If yes, a CFA should be performed instead of a EFA (see one previous comment of reviewer for which authors did not replied “Terwee et al. (2007) also suggest that where a theoretical model exists, confirmatory factor analysis should be used. As this study modelled the adaptation of a pre-existing scale for adults to an adolescent population, and given the authors confidence in the results of the EFA, use of CFA could be justified and would provide further credit to the design of the scale”. I have the same comment, I do not understand why authors did not performed a CFA since a theoretical model already exists”. 

Response 10: We did not think that a theoretical model already exists as factor structures of the original GTS-QOL and the French version of the GTS-QOL for adults were different. The same applies for adolescents with two different factor structures of the English and Italian versions of the C&A-GTS-QOL. In the framework of a CFA, a hypothesized model is established and the CFA confirm that it provides a good fit to the observed data. This makes CFA a method suited for validation of instruments with a predefined measurement model. Or, given previous studies on the GTS-QOL for adults or adapted for adolescents that found different factor structures, such hypothesized model seems lacking to us. Moreover, the perception and definition of HRQoL may vary from adults to adolescents and could impact the factorial structure of the scale. So we preferred to conduct an EFA to explore the possible underlying factor structure of a set of observed variables without imposing a preconceived structure on the outcome. 

Another reason to not have performed a CFA is the sample size of our study, which is a major inconvenience to conduct a CFA. Anthoine et al. 1 reported that recommendations for the sample size when conducting a CFA are disparate, ranging from 150 to 1000 subjects. The large sample size is a more crucial requirement in CFA compared to EFA 2. 

1 Anthoine E, Moret L, Regnault A, Sbille V, Hardouin JB. Sample size used to validate a scale: a review of publications on newly-developed patient reported outcomes measures. Health and Quality of Life Outcomes. 2014;12:176.

2 Kyriazos TA. Applied Psychometrics: Sample Size and Sample Power Considerations in Factor Analysis (EFA, CFA) and SEM in General. Psychology. 2018; 9:2207-2230.

---

## [Editor Report · Decision Letter 3]

11 Nov 2022

PONE-D-22-04149R3The French version of the Gilles de la Tourette Syndrome Quality of Life Scale for adolescents (GTS-QOL-French-Ado): adaptation and psychometric evaluationPLOS ONE

Dear Dr. Jalenques,

Thank you for submitting your manuscript to PLOS ONE. After careful consideration, we feel that it has merit but does not fully meet PLOS ONE’s publication criteria as it currently stands. Therefore, we invite you to submit a revised version of the manuscript that addresses the points raised during the review process.

We look forward to receiving your revised manuscript.

Kind regards,

Charlotte Beaudart

Academic Editor

PLOS ONE

Journal Requirements:

Additional Editor Comments:

I thank the authors who responded correctly to my comments.

I would be able to accept the publication of the manuscript once the authors will have reported more precisely in the manuscript their response to my comments n° 5 and 10. It is appreciated to provide me explicit answers but this should be reported transparently in the manuscript as well.

---

## [Author Response · Author response to Decision Letter 3]

14 Nov 2022

To the editor

Journal Requirements:

Response: Two references have been added (numbered 51 and 52).

I would be able to accept the publication of the manuscript once the authors will have reported more precisely in the manuscript their response to my comments n° 5 and 10. It is appreciated to provide me explicit answers but this should be reported transparently in the manuscript as well.

Response: We have reported in the manuscript our responses to the previous comments n°5 (about the sample size of respondents for test-retest reliability) and n°10 (about our choice to not perform a confirmatory factor analysis).

Responses to comment n°5 were reported under Reliability subheading in the Results section and in the Discussion section. 

Responses to comment n°10 were reported in the Discussion section.

---

## [Editor Report · Decision Letter 4]

16 Nov 2022

The French version of the Gilles de la Tourette Syndrome Quality of Life Scale for adolescents (GTS-QOL-French-Ado): adaptation and psychometric evaluation

PONE-D-22-04149R4

Dear Dr. Jalenques,

We’re pleased to inform you that your manuscript has been judged scientifically suitable for publication and will be formally accepted for publication once it meets all outstanding technical requirements.

Kind regards,

Charlotte Beaudart

Academic Editor

PLOS ONE

Additional Editor Comments (optional):

Thank you for providing satisfactory responses to my questions. 
---

## [Editor Report · Acceptance letter]

18 Nov 2022

PONE-D-22-04149R4 

The French version of the Gilles de la Tourette Syndrome Quality of Life Scale for adolescents (GTS-QOL-French-Ado): adaptation and psychometric evaluation 

Dear Dr. Jalenques:

I'm pleased to inform you that your manuscript has been deemed suitable for publication in PLOS ONE. Congratulations! Your manuscript is now with our production department. 

Kind regards, 

on behalf of

Dr. Charlotte Beaudart 

Academic Editor

PLOS ONE